# Opposing roles of pseudokinases NRBP1 and NRBP2 in regulating L1 retrotransposition

Wei Yang[1,2,11], Shaobo Cong[1,11], Ruoyao Li [1], Jennifer Schwarz [3,4], Thilo Schulze[5], Raban A. Gevelhoff [1,6], Xinyan Chen [1,6], Sara Ullrich[1], Kristina Falkenstein[1], Denis Ott[1], Pia Eixmann[1], Angelica Trentino[1], Antje Thien[1], Thierry Heidmann[7], Ekkehard Schulze [1], Bettina Warscheid [3,8], Ralf Baumeister [1,6,9,10] ✉ & Wenjing Qi [1] ✉

Gene duplication generates gene paralogs that may undergo diverse fates during evolution, and thus serves as a potent catalyst of biological complexity. Genetic paralogs frequently share redundant functions and may also exhibit antagonistic activities by competing for common interaction partners. Here we show that the gene paralogs *NRBP1* and *NRBP2* oppositely regulate long interspersed nuclear element-1 (L1) retrotransposition, via influencing integrity of the L1 ribonucleoprotein complex. We demonstrate that the opposing roles of NRBP1 and NRBP2 are not results of a competitive mechanism, but rather due to targeting NRBP1 for degradation by NRBP2, probably through heterodimer formation. Moreover, our phylogenetic analysis shows that the regulatory function of NRBP2 may be acquired later during evolution, suggesting that evolutionary pressure has favored this functional fine-tuning of NRBP1. In summary, our findings not only identify NRBP1/2 as L1 regulators and implicate their involvement in human pathogenesis, but also provide a mechanistic insight into the regulatory details arising from gene duplication.

Paralogs, arising from gene duplication events, play a key role in supplying new genetic material for natural selection in evolution. The functional redundancy shared by paralogs serves as a crucial source of genetic robustness[1]. Additionally, paralogs may evolve to new functions, either complementary to or different from those of the original genes[2]. Despite being the result of gene duplication, the existence of antagonistic functions among paralogs has long been documented[3–5]. This antagonism is often conceptualized within a competition model, where the protein encoded by a loss-of-function gene copy competes with that from its functional sister copy in binding to common interaction partners[4]. The persistence and fixation of such an acquired ability, allowing one gene to repress the activity of its sister gene, suggest evolutionary advantages. This phenomenon likely contributes to enhancing the plasticity of complex biological systems, providing an adaptive edge in the dynamic landscape of evolution.

[1]Bioinformatics and Molecular Genetics, Institute of Biology III, Faculty of Biology, Albert-Ludwigs-University Freiburg, Freiburg, Germany. [2]College of Food Science and Engineering, Shandong Agricultural University, Shandong Engineering Research Center of Food Nutrition and Active Health, Taian 271018, People's Republic of China. [3]Biochemistry-Functional Proteomics, Institute of Biology II, Faculty of Biology, Albert-Ludwigs-University Freiburg, Freiburg, Germany. [4]European Molecular Biology Laboratory (EMBL), Heidelberg, Germany. [5]Department of Animal Evolution and Biodiversity, Georg-August-Universität Göttingen, Untere Karspüle 2, Göttingen 37073, Germany. [6]Spemann Graduate School of Biology and Medicine (SGBM), Albert-Ludwigs-University Freiburg, Freiburg 79104, Germany. [7]CNRS UMR 9196, Laboratory of Molecular Physiology and Pathology of Endogenous and Infectious Retroviruses, Gustave Roussy, University Paris-Saclay, Villejuif, France. [8]Biochemistry II, Theodor-Boveri-Institute, Biocenter, University of Würzburg, 97074 Würzburg, Germany. [9]Signalling Research Centers BIOSS and CIBSS, Albert-Ludwigs-University Freiburg, Freiburg 79104, Germany. [10]Center for Biochemistry and Molecular Cell Research, Faculty of Medicine, Albert-Ludwigs-University Freiburg, Freiburg, Germany. [11]These authors contributed equally: Wei Yang, Shaobo Cong. ✉e-mail: baumeister@celegans.de; wenjing.qi@biologie.uni-freiburg.de

Nuclear receptor-binding proteins (NRBPs) are evolutionarily conserved pseudokinases that, despite losing their original function of phosphorylating proteins, can act as allosteric modulators, scaffolds for complex assembly, or as competitive inhibitors in signaling pathways[6]. Although NRBP family members widely exist in metazoan, their molecular functions are not well known. Humans have two *NRBP* paralogs, *NRBP1* and *NRBP2*. NRBP1 protein consists of a conserved kinase-like domain that has lost its kinase activity, nuclear export/import signals (NES/NLS), a BC-binding box, a MLF1-binding region and two predicted nuclear receptor binding (NRB) motifs[7–9]. The BC-binding box has been shown to be responsible for association of NRBP1 with Elongin B/C (ELOB/C)[10]. The MLF1-binding region has been reported to mediate NRBP1 homodimerization which is crucial for assembly of the Elongin B/C containing Cullin-RING E3 ubiquitin ligase complex[11]. The best studied molecular function of NRBP1 is its participation in proteasome-mediated protein degradation. Here, NRBP1 acts as a substrate recognition receptor in the Elongin B/C containing Cullin-RING E3 ubiquitin ligase complex and has been shown to promote Amyloid β production via accelerating BRI2 and BRI3 degradation in neuronal cells[11]. In addition, NRBP1 has been found to promote SALL4 degradation and affects various signaling pathways, including Rac1/Cdc42[12,13]. In these contexts, NRBP1 exhibits either oncogenic or tumor-suppressive properties to influence tumorigenesis and development.

In contrast to NRBP1, the molecular function of NRBP2 is much less well characterized. NRBP2 was initially recognized for its participation in supporting neural progenitor cell survival[14]. In addition, NRBP2 is suggested as a tumor suppressor by suppressing key oncogenic signaling pathways, such as AKT and Mammalian Target of Rapamycin (mTOR) pathways[15,16]. How NRBP2 mechanistically influences signaling transduction or whether it might have a similar molecular function as NRBP1, is currently unknown.

Long interspersed nuclear element-1 (L1) is the only known active autonomous retrotransposon in humans and contributes to about 17% of the human genome[17]. L1 encodes two proteins that play essential roles in successful L1 retrotransposition, an RNA-binding protein open-reading frame 1 protein (ORF1p), and an open-reading frame 2 protein (ORF2p) that possesses endonuclease and reverse transcriptase activities[18–22]. ORF1p and ORF2p act together to enable mobilization of L1 through a "copy and paste" mechanism. After transcription from the genome and translation of ORF1p and ORF2p, L1 mRNA together with ORF1p, ORF2p, and other RNA-binding proteins assemble into ribonucleoparticles (L1 RNPs) in the cytoplasm[23–25]. The L1 RNPs subsequently translocate into the nucleus, followed by reverse transcription and integration into new genomic loci[18,26]. Given that L1 retrotransposition can cause mutations and drive genome instability, its activation is primarily linked to human diseases such as tumors[27]. In addition, both L1 mRNA and L1 cDNA have been shown to activate innate immune response[28–31]. Therefore, L1 activation is linked to inflammation and the onset of autoimmune diseases[32–34]. Host cells have developed several defense mechanisms to prevent deleterious L1 mobilization. Most of the L1 DNA copies are inactive due to mutations, rearrangements, or truncations. In addition, DNA methylation and histone modification silence L1 at the transcriptional level[35]. Furthermore, a variety of ORF1p-associated host factors are reported to restrict L1 via different mechanisms[36,37]. Despite the obvious deleterious consequences of L1 activation, it also has important biological roles under certain circumstances[38]. L1 activity in the germline is considered to be an important source of genetic diversity[39,40]. In addition, the L1 transcript has been reported to function as non-coding regulatory RNA and to actively regulate neuronal development and brain function[41]. Therefore, regulatory mechanisms for both L1 activation and inhibition must have been co-evolved to enable a context- and tissue-specific control of L1.

Here we identify NRBP1 and NRBP2 as regulators of L1 retrotransposition. We show that NRBP1 and NRBP2 interact with L1 ORF1p but exert opposing effects on L1 mobility. Specifically, NRBP1 activates, whereas NRBP2 restricts L1 retrotransposition by influencing the association of L1 mRNA with ORF1p. Moreover, we demonstrate that the restrictive role of NRBP2 is achieved by targeting NRBP1 protein for degradation, probably through heterodimer formation between NRBP1 and NRBP2. Furthermore, NRBP2 knockdown results in activation of innate immune response, which is partially dependent on L1, and NRBP2 expression level displays a negative correlation with rheumatoid arthritis (RA) autoimmune disease. Finally, our phylogenetic analysis shows that *NRBP1/2* emerge as gene duplication products of an ancestral *NRBP* in the early vertebrate lineage. *NRBP1* probably maintains the functions of its ancestral *NRBP*, while *NRBP2* may have obtained additional regulatory roles during its evolution. In summary, this study not only reveals the opposite roles of NRBP1 and NRBP2 in regulating L1 retrotransposition, but also provides a mechanistic insight into how one protein inhibits its paralogous protein via heterodimerization-dependent protein degradation.

## Results

### NRBP1 and NRBP2 interact with L1-encoded ORF1p

As catalytically inactive enzymes, pseudokinases can function as scaffold proteins to mediate protein interactions. Therefore, identifying binding partners of NRBP1 and/or NRBP2 (NRBP1/2) would shed light on the protein complexes wherein NRBP1/2 carry out their functions. Due to the relatively higher expression levels of NRBP1 and NRBP2 in MCF-7 cells (Supplementary Fig. 1a), and the lack of specific antibodies to immunoprecipitate NRBP1 and NRBP2 individually, we immunoprecipitated endogenous NRBP1 and NRBP2 in MCF-7 cells by using an antibody recognizing both pseudokinases (Supplementary Fig. 1b) and identified co-immunoprecipitated proteins via Liquid Chromatography–Mass Spectrometry (LC–MS). In total, we identified 107 proteins as associated with NRBP1/2 (enrichment in NRBP1/2 immunoprecipitation (IP) *vs.* mock IP ≥ 5-fold, $p < 0.05$), including the previously known NRBP1 interactors: ELOB (TCEB2), ELOC (TCEB1), TSC22D1, and TSC22D2 (Fig. 1a and Supplementary Data 1). Gene ontology (GO) analysis of the NRBP1/2 interactors revealed that they are mostly enriched in transcriptional and post-transcriptional regulation of gene expression (Supplementary Data 1). In addition, we noticed that ORF1p, which is encoded by the L1 retrotransposon, was co-immunoprecipitated with NRBP1/2. Moreover, multiple proteins that are known to regulate L1 retrotransposition or associate with ORF1p were also identified as NRBP1/2 interactors (Supplementary Data 1). We next confirmed the interaction of endogenous NRBP1/2 with ORF1p via Co-IP using NRBP1/2 antibody (Fig. 1b). Transfection of MCF-7 cells with Myc-Flag-NRBP1 or Myc-Flag-NRBP2 allowed us to pull down endogenous ORF1p with anti-Flag antibody (Fig. 1c). Flag-tagged ORF1p also enabled pull-down of both NRBP1-Myc and NRBP2-Myc in an RNA-independent manner (Fig. 1d, e). These results together suggest that both NRBP1 and NRBP2 interact with ORF1p and this is not caused by RNA-mediated tethering. Furthermore, we confirmed the association of both NRBP1 and NRBP2 with some known ORF1p interactors that are also identified in our MS analysis, including UPF1, MOV10, G3BP1 and YBX1 (Supplementary Fig. 1c), suggesting that NRBP1 and NRBP2 might be additional components of the previously described L1 RNP complexes that affect L1 mobility.

### NRBP1 and NRBP2 regulate L1 retrotransposition in opposite ways

The formation of the cytosolic L1 RNP complex, initiated by the binding of L1 mRNA to ORF1p, is a crucial prerequisite for L1 retrotransposition. Many regulators of L1 retrotransposition exert their

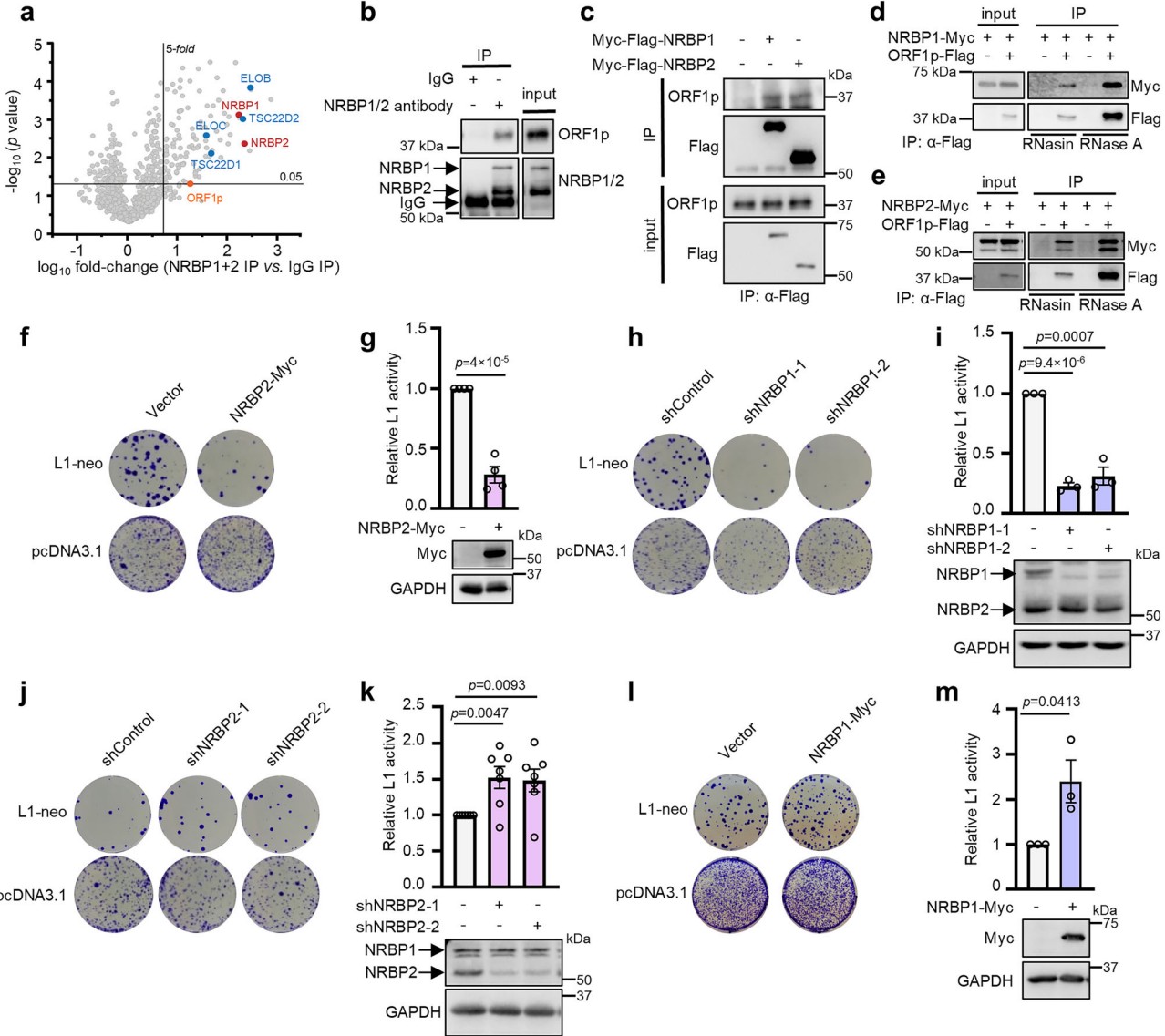

**Fig. 1 | NRBP1 and NRBP2 interact with L1-encoded ORF1p and contrarily regulate L1 retrotransposition. a** Scatter plots showing proteins significantly enriched (fold change ≥ 5, $p < 0.05$) in NRBP1/2 immunoprecipitation (IP) *vs.* IgG in MCF-7 cells. NRBP1 and NRBP2 are shown in red. Previously known NRBP1 interactors are labeled in blue. ORF1p is marked in orange. Statistical significance for protein enrichment was assessed using a one-sided two-sample Student's *t*-test, with a valid value filter applied to at least two out of three replicates. No multiple testing correction was applied. **b** Endogenous ORF1p is co-immunoprecipitated with NRBP1 and/or NRBP2. An antibody recognizing both NRBP1 and NRBP2 was used to pull down endogenous NRBP1 and NRBP2 in MCF-7 cells. Co-precipitated ORF1p was detected by using ORF1p antibody. $n = 3$ biological replicates with similar results. Uncropped blots in Source Data. **c** Endogenous ORF1p is co-immunoprecipitated with both NRBP1 and NRBP2. MCF-7 cells were transfected with either Myc-Flag-tagged NRBP1 or Myc-Flag-tagged NRBP2. The cell lysates

were incubated with Flag antibody and the immunoprecipitated proteins were detected by Western blot. $n = 2$ biological replicates with similar results. Uncropped blots in Source Data. **d, e** NRBP1 (**d**) and NRBP2 (**e**) are co-immunoprecipitated with ORF1p with or without RNA in HeLa cells. RNasin and RNase A were used to protect or digest RNA in the cell lysate. $n = 3$ biological replicates with similar results. Uncropped blots in Source Data. **f-m**, NRBP1 activates and NRBP2 inhibits L1 retrotransposition. Representative colony formation assays are shown in (**f**), (**h**), (**j**) and (**l**). Quantification and representative Western blots are shown in (**g**), (**i**), (**k**), and (**m**). Data: mean ± SEM; two-sided unpaired t-test without multiple comparison adjustment. $n = 4$ (**f, g**), 3 (**h, i, l, m**) biological replicates. Due to the small increase and high variability in L1 activity upon NRBP2 knockdown, seven biological replicates were performed for the assay in (**j**) and (**k**). Retrotransposition activity was normalized to pcDNA3.1 control to account for possible cytotoxic effects of NRBP1/2 manipulation.

---

regulatory roles through interacting with ORF1p[36,37,42,43]. We asked whether NRBP1 or NRBP2 might influence L1 retrotransposition. Successful retrotransposition results in cellular resistance to G418 in cells expressing a L1-neo reporter[44]. Since HeLa cells are commonly used to monitor L1 retrotransposition[45], we co-transfected them with the L1-neo reporter or pcDNA3.1 (which encodes the G418 resistance gene), together with NRBP1-myc or NRBP2-myc expression constructs, followed by G418 selection. We also transfected the L1-neo reporter or pcDNA3.1 into the NRBP1 or NRBP2 shRNA knockdown HeLa cell lines.

The relative retrotransposition activity was calculated by dividing the number of G418-resistant colonies in L1-neo-transfected cells by those in pcDNA3.1-transfected control cells, thereby controlling for potential cytotoxic effects caused by overexpression or knockdown[46]. The results of the colony assays showed that either NRBP2 overexpression or NRBP1 knockdown strongly reduced L1 retrotransposition while NRBP2 knockdown or NRBP1 overexpression moderately increased L1 activity (Fig. 1f−m). Furthermore, MTT assays confirmed normal cell viability upon NRBP1 knockdown, and only a slight reduction in cell

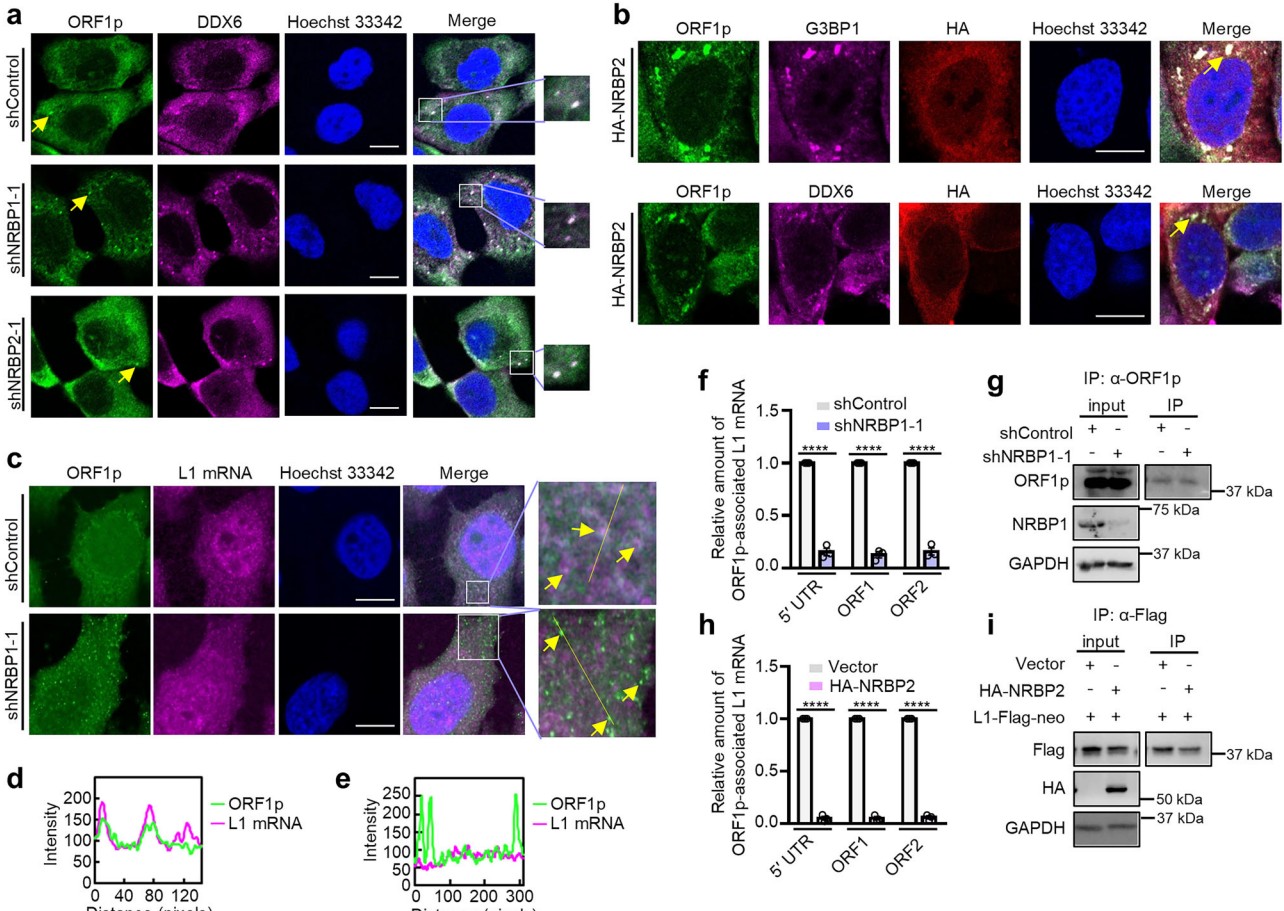

**Fig. 2 | NRBP1 and NRBP2 contrarily regulate ORF1p and L1 mRNA association.**
**a** ORF1p (green) forms cytoplasmic foci co-localizing with DDX6 (magenta) in MCF-7 cells, and their formation is enhanced upon NRBP1 knockdown. Nuclei: blue. Yellow arrows indicate ORF1p foci. Scale bar 10 μm. $n = 3$ biological replicates with similar results. **b** NRBP2 (red) overexpression induces ORF1p (green) foci co-localizing with G3BP1 (magenta) and DDX6 (magenta) in MCF-7 cells. Nuclei: blue. Scale bar 10 μm. $n = 2$ biological replicates with similar results. **c** L1 mRNA is not enriched in ORF1p foci upon NRBP1 knockdown in MCF-7 cells. Immuno-fluorescence (ORF1p, green) and smFISH (L1 mRNA, magenta) are shown. Arrows (top) indicate partial co-localization of ORF1p and L1 mRNA; arrows (bottom) indicate ORF1p puncta lacking L1 mRNA enrichment. Nuclei: blue. Scale bar 10 μm. $n = 3$ biological replicates with similar results. **d** Line profile (yellow line in **c**, top) shows partial co-localization between L1 mRNA and ORF1p in shControl cells. Pearson's $r = 0.74$. **e** Line profile (yellow line in **c**, bottom) reveals no enrichment of

L1 mRNA in ORF1p foci upon NRBP1 knockdown. Pearson's $r = -0.23$. **f** ORF1p RIP-qPCR quantifying endogenous ORF1p-associated L1 mRNA upon NRBP1 knock-down in MCF-7 cells. Primer pairs target L1 5′ UTR, ORF1, ORF2. $n = 3$ biological replicates. Data: mean ± SEM. Two-sided unpaired t-test; no multiple comparison adjustment. $p = 6.25 \times 10^{-6}$ (5′ UTR), $8.7 \times 10^{-6}$ (ORF1), $3.6 \times 10^{-7}$ (ORF2). $****$ $p < 0.0001$. **g** Western blot showing NRBP1 knockdown efficiency and ORF1p IP in (**f**). Uncropped blots in Source Data. **h** NRBP2 overexpression prevents the inter-action between ORF1p and L1 mRNA. RIP-qPCR was carried out to quantify Flag-tagged ORF1p co-immunoprecipitated L1 mRNA levels in HeLa cells. $n = 3$ biological replicates. Data: mean ± SEM. Two-sided unpaired t-test; no multiple comparison correction. $p = 6.83 \times 10^{-8}$ (5′ UTR), $3.7 \times 10^{-9}$ (ORF1), $6.41 \times 10^{-7}$ (ORF2). $****$ $p < 0.0001$. **i** Western blot assesses efficiency of HA-NRBP2 transfection and ORF1p IP for the experiments in (**h**). Uncropped blots in Source Data.

growth by NRBP2 overexpression that was much less pronounced than the observed suppression of L1 mobility (Supplementary Fig. 1d, e), suggesting that the reduction in L1 activity was not due to cytotoxicity. These observations together suggest NRBP1 as a positive and NRBP2 as a negative regulator of L1 retrotransposon.

## NRBP1 and NRBP2 contrarily regulate ORF1p and L1 mRNA association

Altering subcellular localization of L1 RNP complexes can affect L1 retrotransposition[47,48]. Since both NRBP1 and NRBP2 interact with ORF1p and regulate L1 activity, we asked whether they could affect subcellular localization of ORF1p. As MCF-7 cells have a higher endo-genous ORF1p expression level compared to HeLa cells (Supplementary Fig. 2a), we examined localization of endogenous ORF1p in MCF-7 cells. ORF1p antibody staining showed a much stronger signal than the IgG control, indicating the antibody's specificity (Supplementary Fig. 2b). Previous studies have demonstrated that the subcellular

localization of endogenous ORF1p varies significantly across different cell types. In some cases, ORF1p forms cytoplasmic puncta that colo-calize with processing bodies (P-bodies) or stress granules (SGs). However, in other cell types, such puncta are not observed[49–52]. We observed that ORF1p in MCF-7 cells exhibited a predominantly diffuse distribution, accompanied with enrichment in a few small punctate structures in the cytoplasm (Fig. 2a and Supplementary Fig. 2b–e). The number of ORF1p containing puncta increased upon knockdown of NRBP1, while NRBP2 knockdown did not significantly affect number of ORF1p puncta (Fig. 2a and Supplementary Fig. 2c). In addition, we found that the ORF1p containing puncta in both control and NRBP1/2 knockdown cells were mostly positive for the P-body marker DDX6[53], but rarely for the SG markers G3BP1 or TIAR (Fig. 2a and Supplementary Fig. 2d, e). These data together indicate that NRBP1 probably prevents translocation of ORF1p into P-bodies. As NRBP1 knockdown failed to cause translocation of an RNA-binding deficient ORF1p mutant (ORF1p N157A/R159A) into cytoplasmic puncta

(Supplementary Fig. 2f)[43,50], NRBP1 may only affect subcellular localization of RNA-associating ORF1p.

We next checked the influence of NRBP2 and NRBP1 overexpression on L1 ORF1p localization. Overexpression of HA-NRBP2 in MCF-7 cells resulted in an enrichment of endogenous ORF1p in puncta positive for both DDX6 and G3BP1 (Fig. 2b). As the retrotransposition assay was performed in HeLa cells, the influence of HA-tagged NRBP1/2 on ORF1p from transfected L1-Flag-neo plasmids was tested in these cells. Flag-tagged ORF1p in HeLa cells was mostly diffusely localized in the cytoplasm, with a few enrichments of cytoplasmic puncta, which did not colocalize with G3BP1 (Supplementary Fig. 3a). Transfection of HA-NRBP2, but not HA-NRBP1, led to the translocation of ORF1p into large, irregular, G3BP1-positive puncta, despite HA-NRBP2 itself not being enriched in these structures. Moreover, NRBP1 and ORF1p exhibited a similar distribution pattern (Supplementary Fig. 3a). Given their RNA-independent interaction, NRBP1 and ORF1p may colocalize. As transfected HA-NRBP1 in HeLa cells displayed reticular pattern in the perinuclear region, consistent with the previous report[54], we isolated rough ER fraction from HeLa cells and could detect both NRBP1 and NRBP2 (Supplementary Fig. 3b). As none of the pseudokinases contain an ER targeting sequence, NRBP1 and NRBP2 might be attached to ER via interacting with other proteins anchored at the ER membrane.

Sequestration of ORF1p and L1 mRNA into SGs has been proposed as an inhibitory mechanism to prevent L1 retrotransposition[47]. However, the causal relationship between SG sequestration of ORF1p and the inhibition of retrotransposition remains a topic of ongoing debate. The paralog proteins G3BP1 and G3BP2 play essential and redundant roles in SG assembly[55]. Since both G3BP1 and G3BP2 were NRBP1/2 interactors according to our MS and Co-IP results (Supplementary Fig. 1c, Supplementary Data 1), we wondered whether NRBP2 might facilitate SG assembly, leading to the subsequent sequestration of ORF1p into SGs and, thus, inactivation of L1. Neither G3BP1 single knockout nor G3BP1 and G3BP2 double knockdown prevented ORF1p translocation into the large puncta induced by NRBP2 overexpression (Supplementary Fig. 3c, d). This is consistent with a previous report showing that transfected ORF1p still formed puncta in G3BP1/2 double knockout cells, suggesting that these ORF1p puncta are either not SGs or specific G3BP1/2 independent SGs[49]. The inhibitory effect of NRBP2 overexpression on L1 retrotransposition was also G3BP1 independent (Supplementary Fig. 3e, f). These data implicate that ORF1p upon NRBP2 overexpression shuttles into certain cellular puncta whose biogenesis is independent of G3BP1/2, and NRBP2 inhibits L1 retrotransposition independent of the G3BP1-mediated SG pathway.

Association of L1 mRNA with ORF1p to form the L1 RNP complex is a prerequisite for its retrotransposition life cycle. We wondered whether NRBP1 could affect L1 RNP complex assembly. We then used single-molecule fluorescence in situ hybridization (smFISH) to detect endogenous L1 mRNA in MCF-7 cells and examined the influence of NRBP1 on subcellular localization of L1 mRNA and ORF1p. Our L1 smFISH probes yielded punctate signal which vanished upon RNase A treatment (Supplementary Fig. 4a). Two other functional probes against *Caenorhabditis elegans* (*C. elegans*) transcripts *fat-7* and *trcs-1*, previously validated in a published study[56], did not give rise to specific signal in our MCF-7 cells (Supplementary Fig. 4b). These data suggest that our L1 smFISH staining assay is specific. In control cells, ORF1p and L1 mRNA were predominantly diffusely distributed throughout the cytoplasm, making it difficult to assess their colocalization (Fig. 2c). However, upon NRBP1 knockdown, the formation of ORF1p-containing puncta became more prominent, yet these structures did not exhibit an enriched signal of L1 mRNA (Fig. 2c). Quantification of the colocalization between ORF1p and L1 mRNA revealed a significant reduction in their association within these puncta (Fig. 2d, e). This observation led to the hypothesis that NRBP1 knockdown might result in dissociation of ORF1p and L1 mRNA. To test this, we used ORF1p antibody

to pull down endogenous ORF1p in MCF-7 cells and quantified the co-precipitated L1 mRNA by RT-qPCR (RIP-qPCR). Three pairs of primers targeting different parts of L1 mRNA were used (5′ UTR, ORF1p-coding region, ORF2p-coding region). We found that NRBP1 knockdown strongly reduced the amount of L1 mRNA co-immunoprecipitated with ORF1p without a significant influence on total L1 mRNA level (Fig. 2f, g and Supplementary Fig. 4c). These observations suggest an impaired binding of ORF1p to L1 mRNA in the absence of NRBP1. We also transfected an NRBP1 expressing plasmid together with L1-Flag-neo reporter and checked the impact of NRBP1 overexpression on ORF1p/L1 mRNA association and L1 mRNA levels. We observed an increased association between ORF1p and L1 mRNA in two of the three biological replicates, although NRBP1 overexpression slightly decreased L1 mRNA levels (Supplementary Fig. 4d–f). However, due to high variability, the result was not statistically significant. The possible reason for these observations is that ORF1p may efficiently bind to L1 mRNA under normal conditions, which makes a further enhancement of this RNA-protein interaction much more difficult than abrogating the interaction. This could also explain why NRBP1 knockdown or NRBP2 overexpression exerts a robust inhibitory effect on L1 retrotransposition, while the activating effect of NRBP1 overexpression or NRBP2 knockdown is rather moderate and shows greater variability (Fig. 1f–m).

Next, we asked whether NRBP2 also affects ORF1p and L1 mRNA association. We found that NRBP2 overexpression resulted in a more than 90% reduction of ORF1p-associated L1 mRNA and about 60% reduction of L1 total mRNA level (Fig. 2h, i and Supplementary Fig. 4g), indicating that NRBP2 may not only interfere with interaction between ORF1p and L1 mRNA, but also negatively affect L1 expression level. In addition, we observed a consistent increase in ORF1p/L1 mRNA association and a slight elevation of L1 mRNA level upon NRBP2 knockdown (Supplementary Fig. 4h–j), the extent of these effects was weak, in line with its mild influence on L1 retrotransposition (Fig. 1j, k).

In summary, our results suggest that NRBP1 and NRBP2 may affect the affinity between ORF1p and L1 mRNA in an opposite manner, thereby exerting antagonistic roles in controlling L1 retrotransposition. NRBP2 may additionally limit L1 mRNA expression level to inhibit L1 mobility.

## NRBP2 negatively regulates NRBP1 to inhibit L1 retrotransposition

Opposing functions of proteins encoded by paralogs have been described, frequently due to competition for interaction with common binding partners[4]. As both NRBP1 and NRBP2 interact with ORF1p, we asked whether the presence of one NRBP might abrogate interaction between ORF1p and the other NRBP. We found that neither NRBP2 nor NRBP1 exerted a significant influence on the affinity of the other NRBP with ORF1p (Supplementary Fig. 5a), arguing against such a model of competition between NRBPs in ORF1p binding. Next, we asked whether one of these NRBPs may regulate L1 by inhibiting their respective counterpart. In such a scenario, the influence of the upstream inhibitor would be nullified in the absence of the downstream factor. We found that NRBP2 failed to inhibit L1 activity upon NRBP1 knockdown (Fig. 3a, b), indicating that NRBP2 might inactivate L1 via inhibiting NRBP1. In addition, we noticed that NRBP2 overexpression reduced the protein level of endogenous NRBP1 in HeLa cells (Fig. 3c). To further investigate the potential relationship between NRBP1 and NRBP2 expression, we performed both siRNA- and doxycycline (DOX)-inducible shRNA-mediated knockdown of either NRBP1 or NRBP2 in HeLa cells, and examined the effect on the expression level of the other protein. NRBP2 knockdown by siRNA moderately increased NRBP1 protein levels without affecting its mRNA expression (Fig. 3d–f and Supplementary Fig. 5b). NRBP1 siRNA knockdown increased both NRBP2 mRNA and protein levels (Fig. 3d and Supplementary

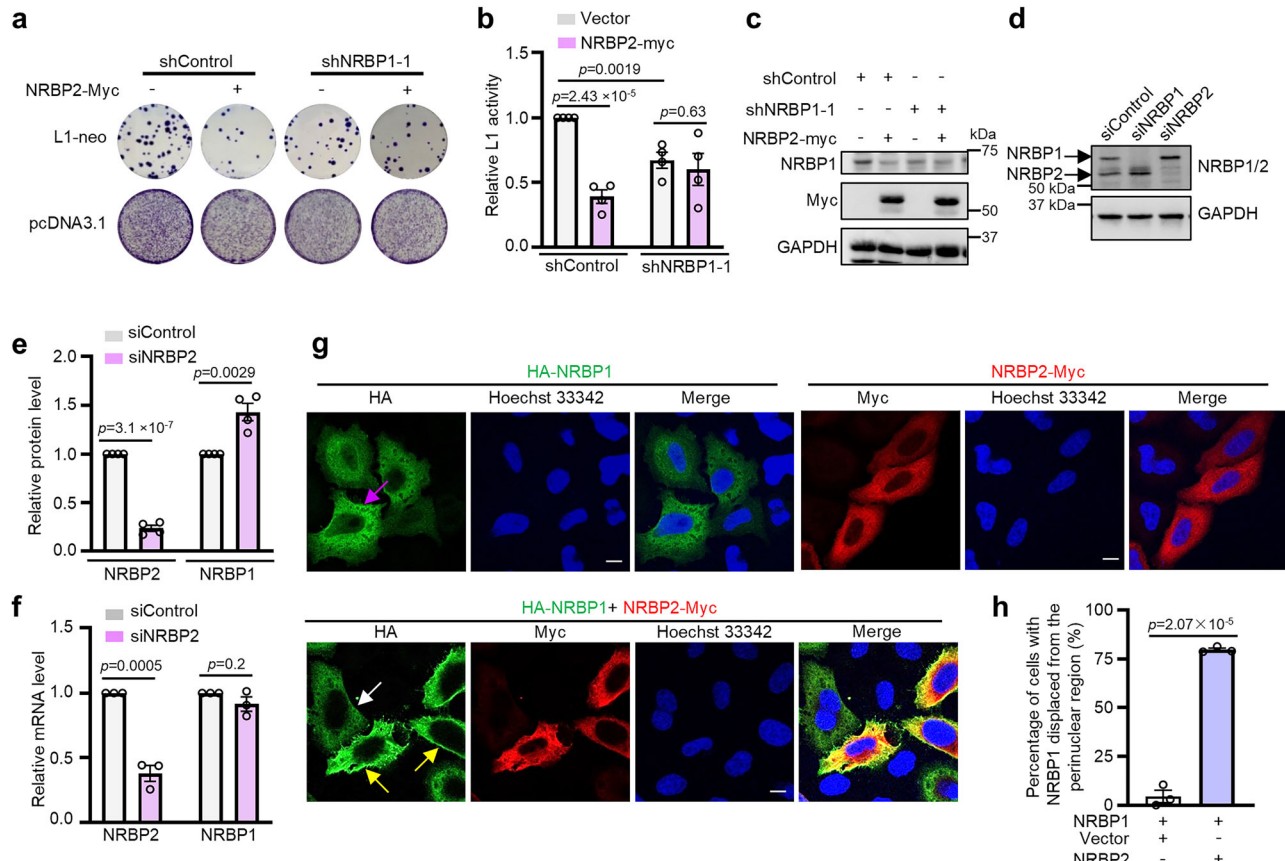

**Fig. 3 | NRBP2 inactivates L1 retrotransposition via NRBP1 inhibition. a** NRBP1 knockdown abolishes inhibitory effect of NRBP2 on L1 retrotransposition. Shown is a representative picture of the colony assay. $n = 4$ biological replicates.
**b** Quantification of L1 retrotransposition activity depicted in (**a**). $n = 4$ biological replicates. Mean ± SEM. Two-sided unpaired t-test; no multiple comparison adjustment. **c** Overexpression of NRBP2-Myc reduces endogenous NRBP1 protein levels. Shown is a representative Western blot using three of the four biological replicates from (**a**). Uncropped blots in Source Data. **d, e** NRBP1 and NRBP2 negatively affect the protein levels of each other in HeLa cells. Shown is one representative Western blot result (**d**). Quantification of the protein levels was shown in (**e**). $n = 4$ biological replicates. Mean ± SEM. Two-sided unpaired t-test; no multiple comparison adjustment. Uncropped blots in Source Data. **f** NRBP2 knockdown does not increase NRBP1 mRNA levels. Shown is quantification of

NRBP1 and NRBP2 mRNA levels normalized to GAPDH in HeLa cells. $n = 3$ biological replicates. Data are mean ± SEM. Two-sided unpaired t-test; no multiple comparison adjustment. **g** Overexpression of NRBP2-Myc (red) results in an enrichment of HA-NRBP1 (green) to the peripheral region of HeLa cells. Yellow arrows point to cells co-expressing HA-NRBP1 and NRBP2-Myc. White arrow indicates a cell only expressing NRBP1. Magenta arrow shows the reticular structure. Nuclei: blue. Scale bar 10 μm. Shown are representative images of three independent experiments. **h** Percentage of cells with NRBP1 displacement from the perinuclear region. Each dot represents an average result of one biological replicate ($n = 3$). The number of cells counted in each replicate is as follows: NRBP1 alone ($n = 57, 34, 14$); NRBP1 and NRBP2 co-transfection ($n = 48, 56, 28$). Data are mean ± SEM. Two-sided unpaired t-test; no multiple comparison adjustment applied.

Fig. 5b–d). Similarly, shRNA-mediated knockdown of NRBP1 led to increased NRBP2 protein levels, while the effect of NRBP2 shRNA on NRBP1 protein levels was variable (Supplementary Fig. 5e–g). This variability could be due to the variation in NRBP2 knockdown efficiency, or may suggest that the regulatory influence of endogenous NRBP2 on NRBP1 might be context-dependent and relatively subtle under the experimental condition. These observations together implicate that NRBP1 and NRBP2 may inhibit the expression level of their respective paralogs, but via distinct mechanisms: while NRBP1 probably affects transcription or mRNA stability of NRBP2, NRBP2 negatively influences NRBP1 activity at the translational or post-translational level. Moreover, examining protein levels of NRBP1/2 in MCF-7, HEK293T, and HeLa cell lines did not reveal a strictly inverse correlation of these two proteins (Supplementary Fig. 1a).

A frequently observed mechanism of post-translational inhibition involves redirecting a protein to a different subcellular location, so we asked whether NRBP2 can alter NRBP1 subcellular distribution. Although both NRBP1 and NRBP2 were detected at the rough ER, only overexpressed NRBP1 exhibited a reticular staining pattern, whereas overexpressed NRBP2 showed diffuse localization throughout the

cytoplasm (Fig. 3g). In cells co-transfected with both HA-NRBP1 and NRBP2-Myc, HA-NRBP1 became more enriched in the peripheral region, while NRBP2-Myc did not show obvious alteration of its subcellular localization (Fig. 3g, h). In summary, our results suggest that NRBP2 not only reduces protein level, but also influences subcellular localization of NRBP1.

## The C-terminal halves of both NRBP2 and NRBP1 negatively regulate L1 retrotransposition

Differences in the amino acid sequences, and thus structure, of NRBP2 should account for its inhibitory activity on NRBP1. We therefore compared these two closely related proteins. Alignment of the protein sequences revealed that NRBP1 and NRBP2 share 55.7% amino acid identity (Supplementary Fig. 6). In addition, most of the predicted structures of NRBP1 are conserved in NRBP2. We noticed that NRBP1 has a longer N-terminus (1-43 aa) than NRBP2, which was recognized by the Segmasker algorithm and AlphaFold as unstructured low complexity region (LCR) (Supplementary Figs. 6 and 7a, b)[57–59]. We asked whether this LCR might account for the functional difference between NRBP1 and NRBP2. To answer this question, we generated an NRBP1

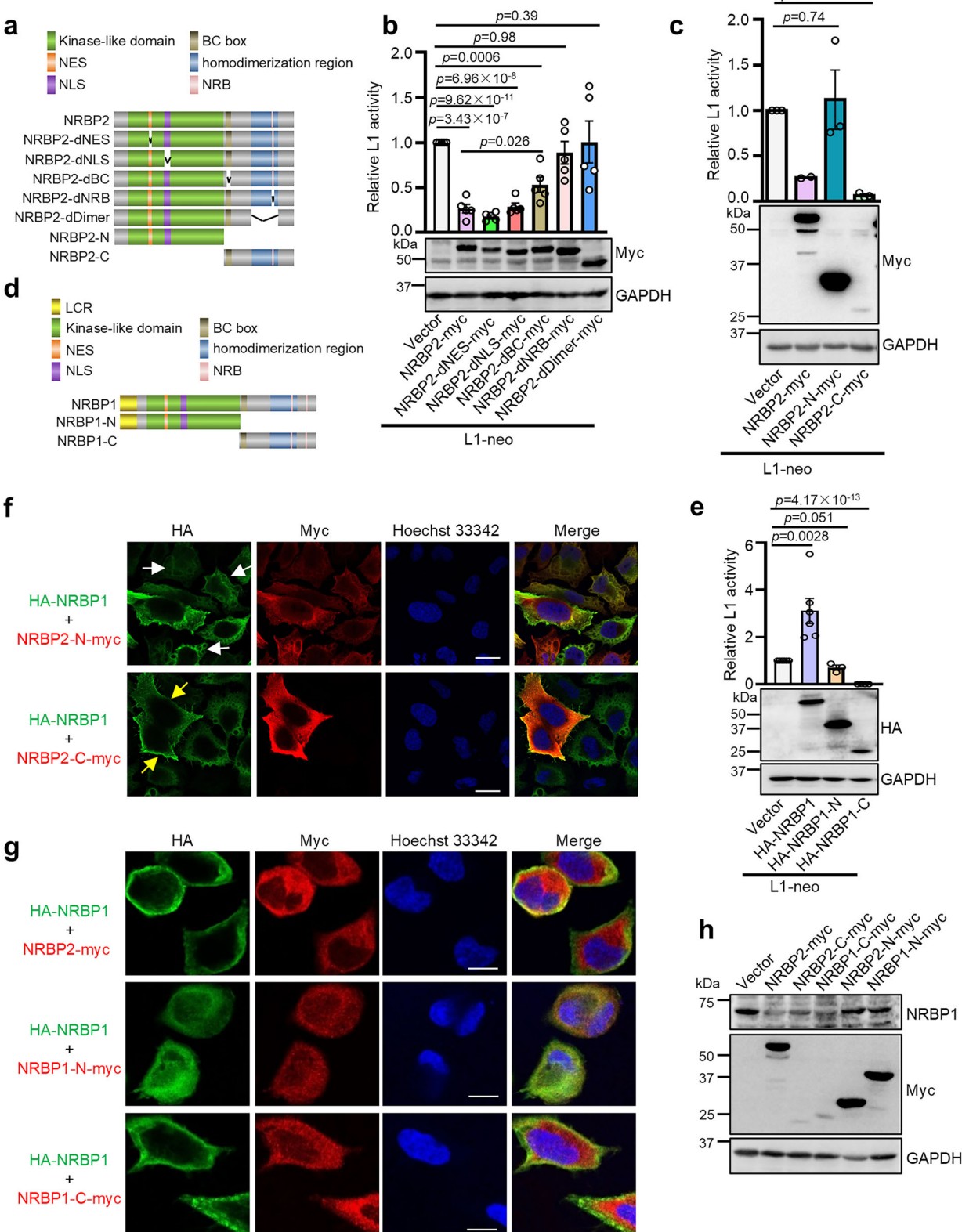

mutant without the LCR (ΔLCR-NRBP1) and a chimeric LCR-NRBP2 mutant by fusing the LCR of NRBP1 to the N-terminus of NRBP2, and tested their impacts on L1 retrotransposition (Supplementary Fig. 7c). We found that these mutants still behaved similarly as their wild-type counterparts (Supplementary Fig. 7d, e), suggesting that the N-terminal LCR does not play a critical role in discriminating NRBP1 from NRBP2.

We reasoned that elucidating critical regions of NRBP1/2 in L1 regulation would be indicative to answer how NRBP2 might inhibit NRBP1. We generated different NRBP2 mutant proteins via eliminating certain domains or motifs and tested their regulatory roles on L1 (Fig. 4a). While NRBP2 mutants lacking the proposed nuclear export signal (NES) or nuclear localization signal (NLS) could still inhibit L1, eliminating the proposed NRB motif (dNRB) or the dimerization region

**Fig. 4 | The C-terminal halves of NRBP2 and NRBP1 inhibit L1 retrotransposition and drive full-length NRBP1 to the peripheral region of cells. a** Schematic diagrams of NRBP2 and its mutants. NES: nuclear export signal; NLS: nuclear localization signal; BC box: Elongin BC-binding motif; NRB: nuclear receptor-binding motif. **b** The NRB and dimerization region are required for NRBP2 to inhibit L1 retrotransposition. Quantification of L1 activity is shown (top). $n = 5$ biological replicates. Data are mean ± SEM. Two-sided unpaired t-test; no multiple comparison adjustment. One representative Western blot confirming expression of NRBP2 variants is shown (bottom). **c** The C-terminal half of NRBP2 is necessary and sufficient to inhibit L1. Shown is quantification of L1 activity (top) and one representative Western blot (bottom). $n = 2$ biological replicates for full-length NRBP2 (no statistical test performed). For NRBP2-N and NRBP2-C, $n = 3$ biological replicates. Data are mean ± SEM. Two-sided unpaired t-test; no multiple comparison adjustment. **d** Schematic diagram of NRBP1 and the NRBP1 halves used in this study. LCR: low complexity region; NES: nuclear export signal; NLS: nuclear localization signal;

BC box: Elongin BC-binding motif; NRB: nuclear receptor-binding motif. **e** The C-terminal half of NRBP1 functions oppositely to the full-length NRBP1 and inhibits L1. Shown is the quantification of L1 activity upon expression of the respective NRBP1 variants. For NRBP1, NRBP1-N, and NRBP1-C, the numbers of biological replicates are 6, 3, and 4, respectively. Data are mean ± SEM. Two-sided unpaired t-test; no multiple comparison adjustment. **f, g** The C-terminal halves of NRBP2 (**f**, red) and NRBP1 (**g**, red) promote enrichment of full-length NRBP1 (green) to the peripheral region of HeLa cells. White arrows in (**f**) indicate NRBP1 localization with NRBP2-N expression; yellow arrows indicate cells with enrichment of NRBP1 to the peripheral region upon NRBP2-C overexpression. Nuclei: blue. Scale bar 10 μm. Shown are representative images of three independent experiments. **h** Overexpressing NRBP2 and C-terminal half of NRBP1 or NRBP2 reduces endogenous NRBP1 protein level in HeLa cells. Shown is a representative Western blot from three independent experiments. Uncropped blots in Source Data.

---

(dDimer) abolished the inhibitory effect of NRBP2. A variant of NRBP2 lacking the Elongin BC-binding motif (dBC) displayed a partial loss of function (Fig. 4b and Supplementary Fig. 7f). As BC-binding, NRB, and dimerization motifs are all localized in the C-terminal half of the NRBP2 protein, we wondered whether this region might exert the inhibitory role. We next tested whether overexpression of N-terminal or C-terminal halves of NRBP2 (NRBP2-N and NRBP2-C, illustrated in Fig. 4a) alone could affect L1 activity. NRBP2-C conferred an even stronger suppressive effect on L1 retrotransposition than wild-type NRBP2, despite its significantly lower expression level. In contrast, NRBP2-N failed to inhibit L1 (Fig. 4c and Supplementary Fig. 7g). These data together suggest that C-terminal half of NRBP2 is necessary and sufficient to inhibit L1. We next asked whether loss of inhibition in NRBP2-N is caused by its loss of ORF1p binding. We found both NRBP2-N and NRBP2-C could be co-immunoprecipitated with ORF1p (Supplementary Fig. 7h), implicating that NRBP2 might interact directly or indirectly with ORF1p through multiple interfaces, and interaction alone is not sufficient for NRBP2-N to restrict L1 retrotransposition.

Since NRBP1 and NRBP2 have opposing activities in L1 regulation, we wondered whether the C-terminal halves of these two proteins are sufficient for these divergent functions. We constructed NRBP1-N and NRBP1-C halves (Fig. 4d) and found that NRBP1-N, as expected, did not show any effect on L1 retrotransposition. To our surprise, NRBP1-C strongly repressed L1 retrotransposition, behaving similarly as NRBP2-C but oppositely to the full-length NRBP1 (Fig. 4e and Supplementary Fig. 7i). These results indicate that the presence of the N-terminal half may inhibit the functionality of the C-terminal half of NRBP1.

Next, we tested whether NRBP2-C or NRBP1-C, which both inhibit L1 activity, would behave similarly as full-length NRBP2 to regulate protein level and subcellular distribution of full-length NRBP1. Transfection of NRBP2-C-Myc or NRBP1-C-Myc resulted in an enrichment of full-length HA-NRBP1 in the cell periphery and a decrease of NRBP1 protein level (Fig. 4f–h and Supplementary Fig. 8), similar as full-length NRBP2. In contrast, overexpression of NRBP2-N-Myc or NRBP1-N-Myc, which alone did not affect L1 activity, altered neither subcellular distribution nor protein level of NRBP1 (Fig. 4f–h). All together, these data suggest that the isolated C-terminal halves of NRBP1 and NRBP2 exert a similar function as NRBP2 to inhibit full-length NRBP1, thereby preventing L1 retrotransposition.

## NRBP2 accelerates degradation of NRBP1 protein via a proteasome-independent mechanism

Our results so far demonstrate that NRBP2 not only negatively affects NRBP1 protein level without affecting its mRNA level, but also promotes a distribution of NRBP1 to the cell periphery. There are two scenarios of how NRBP2 could accomplish this NRBP1 regulation. In the first model, NRBP1 is redirected to the periphery by NRBP2, where it is functionally inactive. In the second model, degradation of NRBP1 in the nuclear periphery is stimulated by NRBP2, and the peripheral

remaining NRBP1 is insufficient to execute its L1-activating function. To test these two possibilities, we asked whether triggering degradation of NRBP1 in the perinuclear region by NRBP2 could account for the change in both protein level and subcellular localization of NRBP1. Firstly, we found that consistent with its inability to regulate L1, NRBP2-dNRB failed to reduce NRBP1 protein levels (Fig. 5a). Next, we used cycloheximide (CHX) to block protein synthesis and observed that, compared with NRBP2-dNRB mutant, wild-type NRBP2 overexpression led to a significantly faster reduction in the levels of existing NRBP1 proteins in HeLa cells (Fig. 5b, c), indicating an accelerated NRBP1 protein degradation by NRBP2 overexpression. In addition, overexpression of NRBP2 failed to reduce NRBP1 protein level in the presence of proteasome inhibitor MG132 (Fig. 5d and Supplementary Fig. 9a). Similarly, both NRBP1-C and NRBP2-C reduced NRBP1 protein level and this could be blocked with MG132 (Fig. 5d and Supplementary Fig. 9a). In contrast, NRBP1-N and NRBP2-N, which did not affect L1 retrotransposition, had no effect on NRBP1 protein level (Fig. 5d and Supplementary Fig. 9a). Consistently, we found that when NRBP1 was co-expressed with either NRBP2-C or NRBP1-C, it showed a preference of localization in the peripheral region of cells and MG132 partially restored its localization in the perinuclear region (Fig. 5e and Supplementary Fig. 9b, c). These observations together indicate that the predominant peripheral localization of NRBP1, triggered by NRBP2 or the C-terminal halves of both NRBPs, is probably a consequence of NRBP1 degradation in the perinuclear region of the cell.

We further asked whether NRBP2 promotes NRBP1 degradation via the proteasome-mediated pathway. Two other proteasome inhibitors, PS-341 and Epoxomicin, failed to restore reduced NRBP1 protein levels upon NRBP2 overexpression (Fig. 5f, g). In addition, we transfected HEK293T cells with plasmids expressing His-tagged ubiquitin and pulled down all ubiquitinated proteins with His-tag purification resin. NRBP2 overexpression did not increase ubiquitination of NRBP1 proteins (Fig. 5h). These data together argue against the model that NRBP2 stimulates ubiquitination of NRBP1 for its degradation by proteasome. In addition to inhibiting proteasome, MG132 has been shown to impair certain lysosomal proteases and calpains[60]. As neither the calpain inhibitor Calpeptin nor the lysosomal acidification inhibitor Bafilomycin A1 (BafA1) could restore the reduced NRBP1 protein levels upon NRBP2 transfection (Fig. 5i, j), an involvement of calpain or lysosomal proteases is less likely.

NRBP1 is known to act as the substrate recognition factor in the Elongin B/C E3 ubiquitin ligase complex to facilitate protein degradation[11], while a role of NRBP2 in protein turn-over has not been described before. An appealing possibility would be that NRBP2 could trigger degradation of NRBP1 via an Elongin B/C dependent way. However, knockdown of ElOB or ElOC did not prevent NRBP1 degradation upon overexpression of NRBP2 or NRBP1-C, NRBP2-C (Supplementary Fig. 9d, e), suggesting that they target NRBP1 for decay independently of the Elongin B/C E3 ubiquitin ligase complex. This

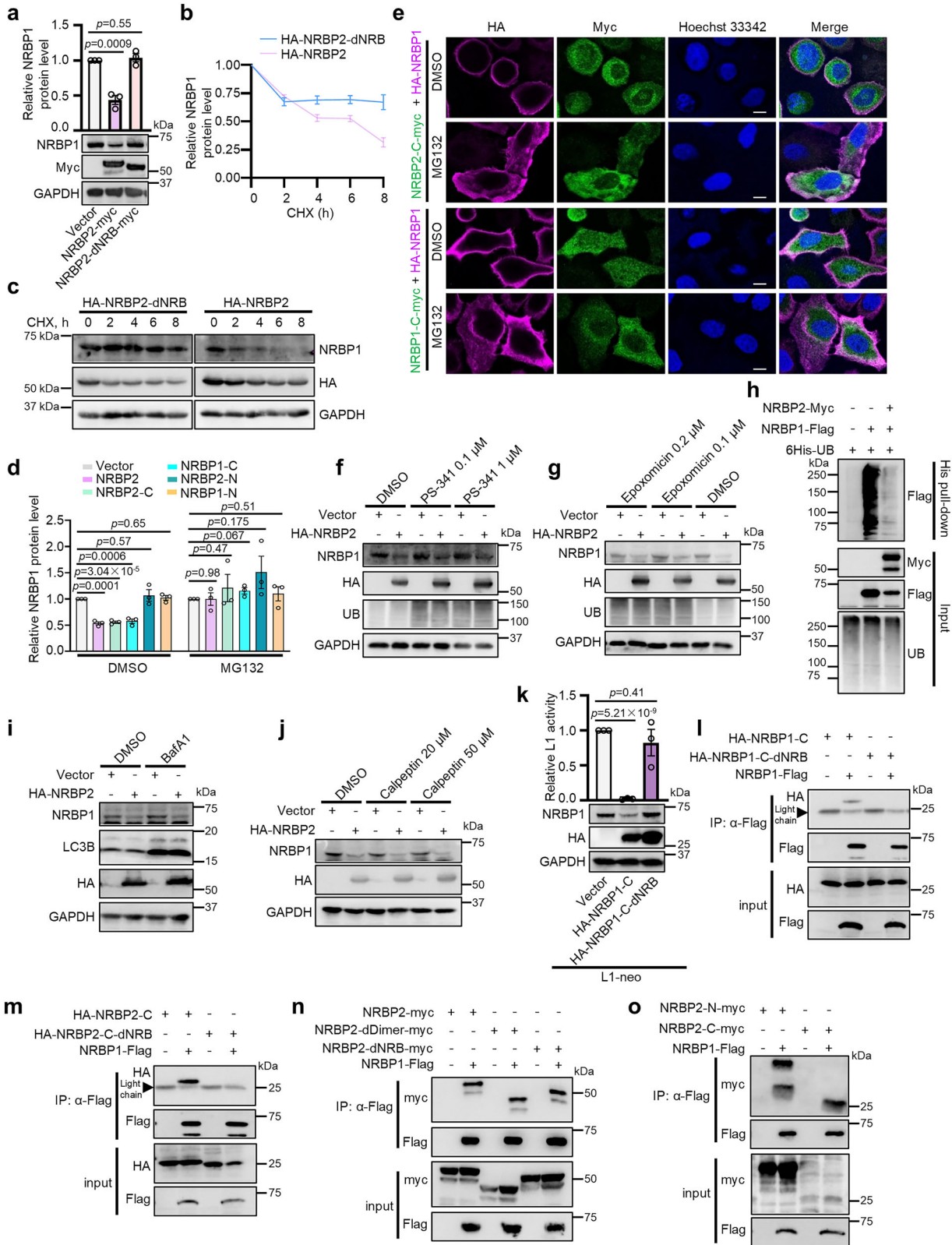

result is also in line with the observations suggesting a proteasome-independent mechanism triggered by NRBP2 to degrade NRBP1.

**Targeting NRBP1 to degradation requires its interaction with the C-terminal half of either NRBP1 or NRBP2**

We further explored how NRBP2 could promote protein degradation of NRBP1. As the functional C-terminal halves of both NRBP1 and

NRBP2 contain the dimerization regions, we wondered whether NRBP2 and NRBP1-C might interact with the full-length NRBP1 to promote its degradation. Similar to NRBP2, NRBP1-C without the two NRB motifs was incapable of decreasing NRBP1 protein level and L1 activity (Fig. 5k and Supplementary Fig. 9f), confirming the significance of the NRB motifs for both NRBP2 and NRBP1-C in regulating NRBP1 protein level and L1 retrotransposition. Moreover, our Co-IP results showed that

**Fig. 5 | NRBP2 promotes proteasome-independent degradation of NRBP1 protein. a** NRB motif in NRBP2 is essential to reduce endogenous NRBP1 protein levels in HeLa cells. Top: quantification normalized to GAPDH; bottom: representative Western blot. $n = 3$ biological replicates. Mean ± SEM. Two-sided unpaired t-test; no multiple comparison adjustment. **b** Wild-type NRBP2, but not the NRB-deletion mutant, accelerates endogenous NRBP1 degradation. HeLa cells were treated with cycloheximide (CHX) for the indicated durations. NRBP1 levels were normalized to GAPDH and set to 1 at 0 h. $n = 3$ biological replicates. Mean ± SEM. **c** Representative Western blot from one replicate in (**b**). **d** MG132 abolishes capability of NRBP2 and the C-terminal halves of NRBP1/2 to reduce NRBP1 protein levels (normalized to GAPDH) in HeLa cells. $n = 3$ biological replicates. Mean ± SEM. Two-sided unpaired t-test; no multiple comparison. **e** MG132 prevents NRBP1 (magenta) peripheral enrichment induced by NRBP1/2 C-terminal halves (green) overexpression. Nuclei: blue. Scale bar 10 μm. **f, g** PS-341 (**f**) and Epoxomicin (**g**) do not block NRBP2-mediated endogenous NRBP1 reduction in HeLa cells. Ubiquitin (UB) confirms inhibitor activities. **h** NRBP2 overexpression does not promote NRBP1 poly-ubiquitination. His-tag pull-down of ubiquitinated proteins followed by Flag immunoblotting. **i, j** BafA1 (**i**) and Calpeptin (**j**) fail to block NRBP2-induced NRBP1 reduction in HeLa cells. LC3B confirms BafA1 activity (**i**). **k** NRB motifs in NRBP1-C are essential to inhibit L1 retrotransposition and reduce NRBP1 levels. Top: L1 activity ($n = 3$ biological replicates); bottom: representative Western blot. Mean ± SEM. Two-sided unpaired t-test; no multiple comparison adjustment. **l, m** Co-IP shows C-terminal halves of NRBP1 (**l**) and NRBP2 (**m**) interact with full-length NRBP1 in an NRB-dependent manner. **n** Lack of NRB and dimerization region does not prevent interaction between NRBP2 and NRBP1. **o** Both the N- and C-terminal halves of NRBP2 interact with NRBP1. For **l–o**, Flag antibody was used to pull down Flag-NRBP1 in HEK293T cells. For **e–j** and **l–o**, biological replicates numbers: $n = 3$ for (**f, g–i, l, m, n, o**); $n = 2$ for (**e, j**). All uncropped blots in Source Data.

both NRBP1-C and NRBP2-C were co-immunoprecipitated with full-length NRBP1 and this required their NRB motifs (Fig. 5l, m). These observations together suggest that the NRB motifs are essential for the C-terminal halves of both NRBPs to bind to full-length NRBP1 and to promote NRBP1 degradation. In contrast to our expectation, NRBP1 interacted with both wild-type NRBP2 and the full-length NRBP2 mutants lacking the dimerization region or the NRB motif (Fig. 5n), implicating an involvement of additional motifs or domains in NRBP2 to NRBP1 interaction. Indeed, the N-terminal half of NRBP2 was also immunoprecipitated with NRBP1 (Fig. 5o). Taken together, our data demonstrated that both N- and C-terminal halves of NRBP2 interact with NRBP1 and the most likely consequence is that they form het-erodimers. However, only the NRB-dependent interaction at the C-termini is necessary and sufficient to trigger NRBP1 degradation.

## Down-regulation of NRBP2 activates inflammatory and type I Interferon pathways

L1 activation provokes innate immune response and stimulates type I interferon (IFN) and inflammatory genes via either L1 mRNA or L1 cDNA[28,29,31,61]. Several negative regulators of L1 have been found to repress innate immune response via L1 inhibition[28,31,43]. Given that NRBP2 also negatively regulates L1 mRNA level (Supplementary Fig. 4g, j), we asked whether NRBP2 has a similar impact on genes involved in the innate immune response. We performed mRNA-seq and checked alteration of gene expression in response to NRBP2 knockdown. The mRNA-seq in HeLa cells revealed that NRBP2 knockdown led to increased expression of 427 genes and reduced expression of 335 genes (fold change ≥1.5, FDR < 0.05, Fig. 6a and Supplementary Data 2). GO analysis showed that genes activated upon NRBP2 knockdown were enriched in innate immune response (Supplementary Data 2). In contrast, NRBP1 knockdown had a less prominent impact on gene expression, resulting in 99 upregulated genes and 54 downregulated genes, without significantly affecting the expression of immune-related genes (Fig. 6b and Supplementary Data 2). Comparing our mRNA-seq result with that from a previous study which investigated transcriptional alteration caused by overexpressing L1[31], we found that 26% of the genes upregulated by NRBP2 knockdown were also activated upon L1 overexpression (Fig. 6c) and these overlapping genes were mostly enriched in immune response (Fig. 6d and Supplementary Data 2). We next investigated whether the upregulated immune-related genes upon NRBP2 knockdown might depend on L1 mRNA or cDNA. Quantitative PCR (qPCR) confirmed the upregulation of five (IL32, MMP19, DDX58, TLR1, IL7R) out of eleven selected immune-responsive genes following NRBP2 knockdown in three additional biological replicates (Fig. 6e). Two other genes (IFIT1 and CCL5) exhibited consistently elevated mRNA levels. However, their activation did not reach statistical significance due to significant variability across the replicates. L1 siRNA knockdown abolished activation of five (IFIT1, DDX58, TLR1, CCL5, IL7R) of these seven

immune-responsive genes by NRBP2 knockdown (Fig. 6e and Supplementary Fig. 10a), suggesting NRBP2 represses innate immune response via both L1-dependent and -independent mechanisms. As blocking reverse transcription with 3TC treatment had no effect on the mRNA levels of any tested immune-responsive genes, irrespective of NRBP2 activity (Supplementary Fig. 10b), increased L1 mRNA level rather than L1 cDNA upon NRBP2 knockdown is more likely to contribute to immune stimulation.

Improper activation of innate immune responses is the cause of some autoimmune diseases and mutations in genes responsible for L1 suppression have often been found in autoimmune diseases[28,30,62,63]. NRBP2 knockdown led to the activation of several L1-dependent immune response genes. In addition, it strongly induced the expression of MMP19—an L1-independent target and a matrix metalloproteinase implicated in extracellular matrix remodeling and the pathogenesis of RA[64]. This result prompted us to explore whether there is an inverse correlation between NRBP2 expression level and occurrence of RA. From the Autoimmune Diseases Explorer database (https://adex.genyo.es/)[65,66], we observed a decreased expression of NRBP2 mRNA level within the synovial membrane samples obtained from RA patients, whereas NRBP1 mRNA level was similar in healthy group and RA patients (Fig. 6f). This correlation hints at a potential protective role of NRBP2 against autoimmune diseases such as RA.

## NRBP1/2 emerged in the early vertebrate lineage by gene duplication and underwent divergent evolution

Invertebrate model organisms, such as *C. elegans* and *Drosophila melanogaster* have a single *NRBP1/2* homolog, whereas humans and mice have a set of two paralogous genes residing on two different chromosomes (Supplementary Fig. 11). Our data so far suggest human NRBP2 as an inhibitor of NRBP1. To understand how this regulatory relationship emerged, we aimed to trace the evolutionary history of this gene duplication event that led to the observed functional differentiation.

We obtained a set of 2065 *NRBP* homologs with Blast searches in the UniProt database using human NRBP1 and NRBP2 as queries. We accepted only proteins that, when compared in a second reverse Blast search against the human proteome, identified either human NRBP1 or NRBP2 as top-ranking hits. When we applied the criteria for the identification of pseudokinases, we learned that all the 2065 *NRBP* homologs encode pseudokinases[67].

Subsequently, we used maximum likelihood to infer the phylogeny of all identified *NRBP* homologs. The resulting phylogenetic analysis recovered 670 *NRBP2* as well as 933 *NRBP1* orthologues as separate monophyletic groups with perfect bootstrap support. The two clades are sister groups, and their branching point was also recovered with perfect bootstrap support, suggesting that the data are highly reliable. The remaining genes are separated from *NRBP1* and *NRBP2* and comprise all invertebrate *NRBP* (Fig. 7a).

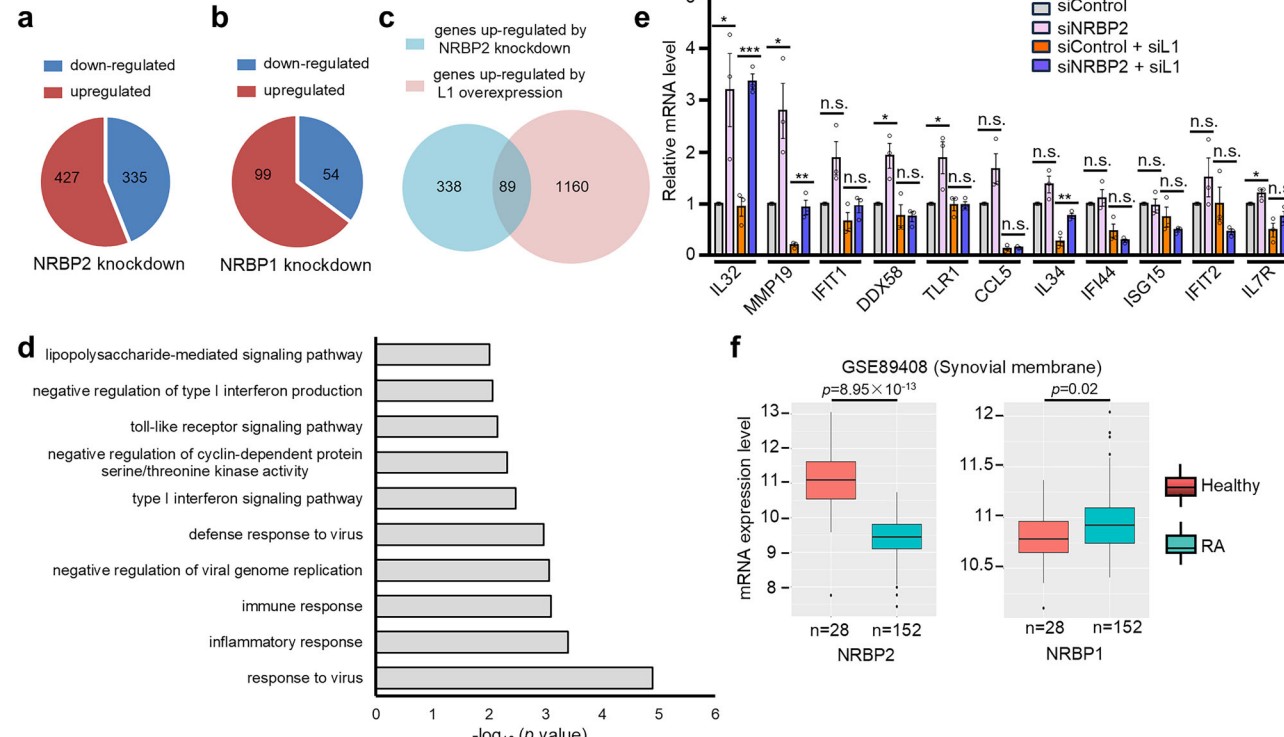

**Fig. 6 | NRBP2 represses innate immune response and NRBP2 mRNA level is negatively correlated with occurrence of RA. a, b** Pie charts illustrating the numbers of genes displaying significantly altered expression levels (fold change ≥ 1.5 and FDR < 0.05) following the knockdown of either NRBP2 (**a**) or NRBP1 (**b**) in HeLa cells. **c** Venn Diagram showing overlapping genes upregulated by NRBP2 knockdown in HeLa cells and genes activated upon L1 overexpression in RPE cells. $p < 2.081 \times 10^{-25}$, calculated using http://nemates.org/MA/progs/overlap_stats.html. **d** GO term analysis for the overlapping genes in (**c**) using DAVID. $p$ values were calculated using a modified Fisher's exact test (EASE score). Terms with $-\log_{10} p > 2$ are shown. No correction for multiple comparisons. **e** NRBP2 knockdown upregulates multiple immune-related genes, partly in an L1-dependent manner. qRT-PCR was performed following NRBP2 knockdown alone or with L1 siRNA in HeLa

cells. $n = 3$ biological replicates. Data are mean ± SEM. Two-sided unpaired t-test; no multiple comparison adjustment. n.s., not significant. * $p < 0.05$, * * $p < 0.01$, * * * $p < 0.001$. $p$ values: IL32 (0.04, 0.0005), MMP19 (0.03, 0.008), IFIT1 (0.053, 0.24), DDX58 (0.02, 0.95), TLR1 (0.04, 0.97), CCL5 (0.08, 0.6), IL34 (0.08, 0.007), IFI44 (0.52, 0.26), ISG15 (0.81, 0.27), IFIT2 (0.24, 0.17), IL7R (0.047, 0.19). **f** mRNA levels of NRBP1 and NRBP2 in samples obtained from both healthy individuals ($n = 28$) and rheumatoid arthritis (RA) patients ($n = 152$). Box plots show the median (center line), the 25th and 75th percentiles (box bounds), and the whiskers extend to the minimum and maximum values, excluding outliers. Outliers are shown as individual points. Two-sided Wilcoxon rank-sum test. No correction for multiple comparisons. Data were retrieved from Autoimmune Diseases Explorer (https://adex.genyo.es).

In addition, the respective phylogenetic lineage information of the 2065 proteins (including human NRBP1 and NRBP2) revealed that *NRBPs* are present in the earliest metazoan clades, including the early-branching Porifera (sponges) and Placozoa. Non-metazoan eukaryotes, such as choanoflagellates, fungi, amoebozoans, plants, bacteria and archaea, do not have *NRBPs* (Supplementary Fig. 12a). While *NRBP1* orthologues are placed within the Vertebrata (vertebrate) category, all the *NRBP2* orthologues belong to the Euteleostomi (bony vertebrates), a sub-clade of the vertebrates (Supplementary Fig. 12b).

Moreover, by analyzing the leaf to root distances in the phylogenetic tree (Fig. 7a), we learned that the proteins comprised in the *NRBP2* clade accumulated considerably more amino acid substitutions than those comprised in the *NRBP1* clade and the average invertebrate *NRBP*. The mammalian *NRBP2* forms a consistent sub-clade, exhibiting an even higher rate of amino acid substitutions (Fig. 7b). This is consistent with our Blast results, in which all invertebrate *NRBPs* produced higher bit scores when compared to human *NRBP1* than to human *NRBP2* (Fig. 7c). This suggests that NRBP1 is more similar to invertebrate NRBP than NRBP2. To test this hypothesis, we transfected a plasmid expressing HPO-11, the sole NRBP in *C. elegans*, into HeLa cells and found that HPO-11 behaved similarly as NRBP1 and stimulated L1 retrotransposition (Fig. 7d and Supplementary Fig. 12c).

Taken together, these observations support a scenario in which *NRBP1/2* originated from a single gene duplication event very early in

the vertebrate lineage. While the more conserved *NRBP1* probably maintained the original *NRBP* functions, *NRBP2* may have evolved to serve specialized functional niches, such as regulating NRBP1 activity by targeting it to degradation.

## Discussion

Gene duplication plays a crucial role in driving functional innovation throughout the course of evolution. Proteins encoded by paralogous genes often exhibit redundant functions, and in some cases, they may even display antagonistic activities. This antagonism is primarily attributed to the competition for common binding partners between the gene products of the functional and loss-of-function copies[4]. A detailed dissection of such a competitive mechanism has explained the antagonistic roles of UPF3A and UPF3B in nonsense-mediated decay machinery[4]. In our study, we present evidences supporting the idea that the antagonistic relationship between paralogs can be established by directing the degradation of the precursor gene product through the action of the later duplicate product. Our findings suggest that this acquired ability to regulate the precursor has been evolutionarily favored, probably through improved flexibility in increasingly complex biological systems.

Instead of competing for ORF1p association, we found the inhibitory role of NRBP2 on L1 relies on the presence of NRBP1. This is achieved by targeting NRBP1 protein for degradation. NRBP1 is known

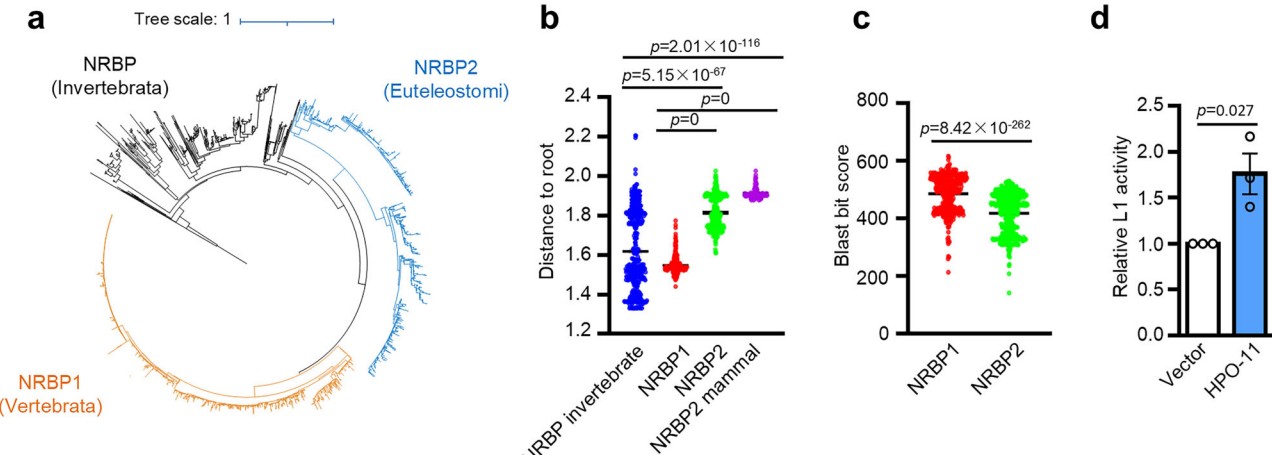

**Fig. 7 | Phylogeny of *NRBP1/2* homologs generated using maximum likelihood.**
**a** Maximum likelihood phylogeny of the IQ-TREE analysis comprising 2,065 quality-filtered NRBP proteins belonging to 663 species and including 417 amino acid positions. Orthologues of human *NRBP2* are recovered as a monophyletic group with perfect bootstrap support (blue). All *NRBP2* belong to the Euteleostomi. Its sister clade contains all orthologues of human *NRBP1* derived from vertebrate taxa (orange) and has perfect bootstrap support as well. The *NRBP* sequence of the sponge *Amphimedon queenslandica* was chosen as outgroup taxon. Branch lengths are amino acid substitutions per site. **b** The distance from leaf to root was determined in (**a**) for all NRBP proteins. The NRBP2 clade accumulated considerably more amino acid substitutions than the NRBP1 clade and the average invertebrate NRBP protein. A consequence of this is that all invertebrate NRBP proteins are more similar to human NRBP1 than they are to human NRBP2. $n = 461$ for NRBP invertebrate, $n = 933$ for NRBP1, $n = 670$ for NRBP2, $n = 203$ for NRBP2 mammal. Statistical significance was assessed using two-sided Welch's t-tests with Benjamini–Hochberg correction for multiple comparisons. **c** Plot of BlastP bit scores resulting from 462 comparisons of invertebrate NRBP either to human NRBP1 or to human NRBP2. Statistical significance was determined using a two-sided paired t-test (n = 462). No multiple comparison adjustment applied. **d** HPO-11 from *C. elegans* enhances L1 activity in HeLa cells. $n = 3$ biological replicates. Data are presented as mean values ± SEM. Two-sided unpaired t-test; no multiple comparison adjustment applied.

to function in the Elongin B/C E3 ubiquitin ligase complex to facilitate the ubiquitination and decay of specific substrates such as BRI2 and BRI3[11]. We showed that, instead, NRBP2 does not increase the ubiquitination of NRBP1, and NRBP1 degradation occurs independently of proteasome, lysosome, or protease calpain. Although the precise mechanism for NRBP1 degradation is unknown yet, our confocal imaging reveals that the degradation of NRBP1 is confined primarily to the perinuclear region of cells, likely due to the presence of the relevant protease in this subcellular area, such as on or near the ER membrane. This suggests that the inhibitory role of NRBP2 on NRBP1 may be limited to NRBP1 functions specifically associated with perinuclear region. However, for functions of NRBP1 occurring at the plasma membrane, such as regulating signal transduction, NRBP2 may not necessarily inhibit NRBP1. Moreover, NRBP1 and NRBP2 did not exhibit a strictly inverse correlation in expression levels across the three cell lines we examined. This could be influenced by additional feedback mechanisms maintaining equilibrium or by variations in the expression of upstream regulators. It is also possible that the inverse relationship between NRBP1 and NRBP2 becomes more pronounced under specific physiological or stress conditions that were not assessed in our study. A broader analysis incorporating more cell types and experimental conditions may provide further insights into this relationship.

NRBP1 is known to form homodimers to function in the Elongin B/C complex. If the ancestral NRBP also forms a homodimer, NRBP2 may have the capacity to form heterodimers with NRBP1 immediately after the gene duplication event, and the capability is possibly conserved during evolution. We propose that the possible heterodimer might render NRBP1 as the substrate of certain protease. Clarifying the underlying degradation mechanism will be crucial for dissecting the interplay between NRBP1/2 in the future. NRBP proteins are named for the presence of NRB motifs. Surprisingly, our study of the NRBP1/2 interactome does not uncover any nuclear receptors as their interactors. Instead, our biochemical assays highlight the significant role of NRB in facilitating both homo- and heterodimerizations of NRBP

proteins. Currently, we cannot definitively determine whether NRB directly participates in dimerization or if its presence is essential for proper protein folding to enable dimerization.

The fact that the C-terminal half of NRBP1 also destabilizes the full-length NRBP1, indicating that the presence of the N-terminal halves of both single NRBP1 molecules in the homodimer may prevent its detection by the protein degradation machinery. In contrast, the N-terminal half of NRBP2 cannot fulfill this protective role, suggesting a perturbation of certain functional motifs during evolution that converted NRBP2 to an inhibitor of NRBP1. Therefore, although the C-terminal half of NRBP2 is necessary and sufficient for NRBP1 degradation, the critical motifs discriminating these two pseudokinases probably exist in their N-terminal halves. Why does the lack of the N-terminal half of NRBP1 in the homodimer trigger its decay? One possible scenario is that the sites for its recognition by the degradation machinery might then be exposed. Or the degradation machinery just recognizes it as a misfolded and defective protein that fails the protein quality control. Therefore, identification of the involved protein degradation machinery would be fundamental to answer these questions. In addition, we anticipate that similar regulatory relationship might also exist among other paralog encoded proteins that function as homo- or heterodimers.

The paralogs *NRBP1* and *NRBP2* in this study encode two highly conserved pseudokinases which, according to our sequence analysis, have lost the ability to phosphorylate proteins at the dawn of their emergence. The wide existence of NRBPs in multicellular animals indicates that they might execute important biological functions. A previous genetic screen by us had revealed the participation of the *C. elegans* NRBP, HPO-11, in tumorigenesis[68]. In addition, numerus recent works suggested regulatory roles of human NRBP1 and NRBP2 in tumor biology, as either tumor suppressor or activator. These reports are, however, mostly descriptive without considerable mechanistic details. Our MS interactome analysis provides an unbiased insight into the potential molecular functions of NRBP1/2. In addition to the known interactors of the Elongin B/C E3 ligase complex, interactors of NRBP1/2

are mostly enriched in RNA-binding proteins and those in transcriptional regulation, implicating that these two pseudokinases might be involved in RNA- and DNA-related processes. Here we suggest NRBP1 and NRBP2 as regulators of L1 retrotransposon, probably through influencing L1 RNP integrity in an antagonistic manner. NRBP2 may additionally repress L1 expression. Remarkably, the emergence of NRBP2 in the Euteleostomi coincided with a switch of the L1 ORF1p from Type I to Type II. It was reported that vertebrates have Type II ORF1p, whereas other animal and plant taxa have Type I ORF1p[69]. However, when we analysed the phylogenetic distribution of Type II ORF1p, which is characterized by PFAM domain PF02994[70], we recognized that Type II ORF1p is actually absent from non-euteleostomian vertebrates. 4,404 of 4,407 (99.9%) of the metazoan reports of PF02994 in the PFAM database belong to the Euteleostomi, and none are found in Agnatha and Chondrichthyes, which also lack *NRBP2* (Supplementary Fig. 12b). The strict correlation of the appearance of *NRBP2* with the switch of L1 transposon to the Type II ORF1p raises the possibility that *NRBP2* might have co-evolved to control this variant of L1.

Although both NRBP1 knockdown and NRBP2 overexpression result in enrichment of ORF1p in certain cytoplasmic foci, these foci are not identical. The ORF1p foci upon NRBP1 knockdown were P-bodies with smaller size while those induced by NRBP2 overexpression were larger and positive for both SG marker G3BP1 and P-body marker DDX6. NRBP2 overexpression may additionally result in certain cellular stress to induce such cellular bodies that contain components of both SGs and P-bodies, including ORF1p, in addition to disassociating ORF1p from L1 mRNA. Although several of our identified NRBP1/2 interactors, such as G3BP1 and MOV10, also trigger translocation of both ORF1p protein and L1 mRNA into SGs and inhibit L1[47,71], NRBP2 does not inhibit L1 activity via SG-mediated mechanism. The data together argue for a quite different strategy used by NRBP1/2 to regulate L1. A previous ORF1p RIP-seq study reported a preference of ORF1p to associate with P-body-enriched mRNAs[72], leading us to speculate whether NRBP1 might enhance ORF1p's affinity to L1 mRNA over P-body-enriched mRNAs or whether NRBP1 could inhibit P-body assembly, thereby reducing the ORF1p protein levels in P-bodies.

The activation of retrotransposons not only induces genome instability, influencing tumor development, but also triggers an innate immune response, contributing to autoimmune diseases. Both L1 cDNA and mRNA have been reported to be triggers of the immune response[28,29,31,43]. Our data support the involvement of L1 mRNA, but not L1 cDNA, in the activation of immune response following NRBP2 knockdown in HeLa cells, as L1 knockdown −but not inhibition of reverse transcription−reduced transcript levels of immune-responsive genes. Furthermore, our RNA-seq and qPCR results showed that only NRBP2 influenced L1 mRNA levels and modulated innate immune activity, suggesting that NRBP2's immune-regulatory role may be independent of NRBP1. To date, NRBP2 has primarily been recognized as a tumor suppressor[15,16]. However, our correlation analysis indicates a negative correlation between NRBP2 with occurrence of autoimmune diseases, and this should be further validated in cellular models better suited for studying immune responses and in clinically relevant samples.

Our finding reveals an antagonistic regulation of L1 retrotransposon by NRBP1 and NRBP2, holding potential clinical implications. Specifically, diseases arising from aberrant L1 activation might be treated with NRBP1 inhibitors or NRBP2 activators. In summary, our discovery that NRBP2 targets NRBP1 for degradation offers profound insights into evolutionary biology, retrotransposon regulation, and therapeutic implications in medicine.

## Methods

### Cell culture
HeLa alpha Kyoto[73] and HEK293T (CRL-3216, ATCC) were cultivated in Dulbecco's modified Eagle's medium (DMEM) with 4.5 g/L glucose (PAN), 10% fetal bovine serum (FBS, Gibco) and 3 mM L-glutamine (Gibco). MCF-7 cells (ACC115, DSMZ) were cultivated in RPMI 1640 (PAN), supplemented with 10% FBS (Gibco) and 3 mM L-glutamine (Gibco).

### Plasmids
The L1 reporter, pCMV-L1-neo$^{TNF}$ (L1-neo) was generated in[44]. pCMV-L1-Flag-neo$^{TNF}$ (L1-Flag-neo, pBY4215) was generated by inserting an in-frame Flag tag to the C-terminus of ORF1p in L1-neo plasmid. pCMV-L1-ORF1p (N157A/R159A)-Flag-neo$^{TNF}$ (L1-ORF1p-mut-Flag-neo, pBY4213) was constructed based on L1-Flag-neo by introducing two point mutations in ORF1p, substituting asparagine at position 157 with alanine (N157A) and arginine at position 159 with alanine (R159A). To construct HA-NRBP1 (pBY4258), HA-NRBP1-N (pBY4278), HA-NRBP1-C (pBY4282), HA-NRBP1-C-dNRB (pBY4287), HA-NRBP2 (pBY4216), HA-NRBP2-C (pBY4271) and HA-NRBP2-C-dNRB (pBY4288), the corresponding coding sequences (CDS) were amplified by PCR and inserted into pRK5-HA vector with the *EcoRI* / *HindIII* sites. To generate NRBP1-myc (pBY4237), NRBP1-N-myc (pBY4281), NRBP1-C-myc (pBY4283), ΔLCR-NRBP1-myc (pBY4254), NRBP2-myc (pBY4243), NRBP2-N-myc (pBY4245), NRBP2-C-myc (pBY4218), NRBP2-dNES-myc (pBY4233), NRBP2-dNLS-myc (pBY4238), NRBP2-dBC-myc (pBY4244), NRBP2-dNRB-myc (pBY4250), NRBP2-dDimer-myc (pBY4255), LCR-NRBP2-myc (pBY4259) and HPO-11-myc (pBY4306), the individual CDS was amplified with primers containing an in-frame stop codon to terminate the N-terminal HA tag in the pRK5-HA vector and the Myc tag was fused at the C-terminus of the CDS. The PCR products were inserted into the *EcoRI* / *HindIII* sites of pRK5-HA. To generate NRBP1-Flag (pBY4275), NRBP1 CDS was amplified with primers containing an in-frame stop codon to terminate the N-terminal HA tag in the pRK5-HA vector and Flag tag was added at the C-terminus of the CDS. The PCR product was inserted into the *EcoRI* / *HindIII* sites of pRK5-HA. Myc-DDK-NRBP1 (pBY4284) and Myc-DDK-NRBP2 (pBY4285) were generated by inserting NRBP1 and NRBP2 PCR products between the *AscI* / *NotI* sites of the pCMV6-AN-Myc-DDK vector (Origene). ORF1p-Flag (pBY4211) was constructed by inserting ORF1p-Flag PCR product into the *BamHI* / *EcoRI* sites of pcDNA3.1+ plasmid. shControl (RHS4743), shNRBP1-1, shNRBP1-2, shNRBP2-1 and shNRBP2-2 are from Dharmacon. Target sequences are shown in Supplementary Data 3.

### Immunoprecipitation and in-gel digestion for Mass Spectrometry (MS)
MCF-7 shControl cells, transduced with shControl-expressing viruses, were used for the NRBP1/2 interactome analysis, as NRBP2 specific interactors were also investigated via comparing co-immunoprecipitated proteins in the shControl with NRBP1 knockdown cells. The NRBP2-specific interactome is beyond the scope of this work and therefore not shown. To perform the immunoprecipitation, cells were lysed in CHAPS-based buffer containing 40 mM HEPES (pH7.5), 120 mM NaCl, 0.3% CHAPS supplemented with Complete Protease Inhibitor Cocktail (Roche, 11697498001), Phosphatase Inhibitor Cocktail 2 (Sigma, P5726) and Cocktail 3 (Sigma, P0044). Proteins were immunoprecipitated with NRBP1/2 antibody (Proteintech, 21549-1-AP) or normal rabbit IgG (Santa Cruz, sc-2027) coupled with Dynabeads™ Protein G (Thermo Fisher, 10009D). The precipitated proteins were separated by NuPAGE® Novex® 4−12% Bis-Tris gels (Invitrogen, NP0322BOX). For in-gel digestion, proteins were destained using 50% ethanol in 10 mM ammonium bicarbonate. For reduction of disulfide bonds and subsequent alkylation, 5 mM tris (2-carboxyethyl) phosphine (10 min at 60 °C) and 100 mM 2-chloroacetamide (15 min at 37 °C) were used, respectively[74].

### High-performance liquid chromatography and MS
LC-MS analysis was performed either on an Ultimate™ 3000 RSLCnano system coupled to an LTQ Orbitrap XL (two replicates) or a

Velos Orbitrap Elite instrument (one replicate). All instruments are from Thermo Fisher Scientific, Bremen, Germany. On both HPLCs a binary solvent system was used with solvent A consisting of 0.1% formic acid and 4% DMSO and solvent B consisting of 48% methanol, 30% acetonitrile, 0.1% formic acid and 4% DMSO.

The HPLC coupled to the LTQ Orbitrap XL was equipped with two PepMapTM C18 μ-precolumns (ID: 0.3 mm × 5 mm, 5 μm, 300 Å Thermo Fisher Scientific) and an AcclaimTM PepMapTM analytical column (ID: 75 μm × 250 mm, 3 μm, 100 Å, Thermo Fisher Scientific). Samples were washed and concentrated for 5 min with 0.1% tri-fluoroacetic acid on the pre-column. A flow rate of 0.250 μL/min was applied to the analytical column and the following gradient was used: 1% B to 30% B in 34 min, to 45% B in 12 min, to 70% B in 14 min, to 99% B in 5 min, 99% B for 5 min and decreased to 1% B in 1 min. The column was re-equilibrated for 19 min with 1% B.

The HPLC coupled to the Velos Orbitrap Elite instrument was equipped with two nanoEase™ M/Z Symmetry C18 trap columns (100 Å pore size, 5 μm particle size, 20 mm length, 180 μm inner diameter) and a nanoEase™ M/Z HSS C18 T3 analytical column (250 mm length, 75 μm inner diameter, 1.8 μm particle size, 100 Å pore size), all from Waters Corporation, Milford, MA. The trap columns were operated at a flow-rate of 10 μL/min and the analytical column at 300 nL/min. Peptide samples were pre-concentrated on the trap column using 0.1% tri-fluoroacetic acid for 5 min before switching the column in line with the analytical column. Peptides were separated using a multi-step gradient. Over the course of 65 min, the percentage of solvent B increased from 3% to 55%, followed by an increase to 80% B in 5 min. The column was eluted for another 5 min with 80% B before returning to 3% B in 4 min. The column was re-equilibrated for 21 min with 3% B.

The MS instruments were operated with the following parameters: 1.5 kV spray voltage, 200 °C capillary temperature. Orbitrap mass range on both instruments m/z 370 to 1700. For the LTQ Orbitrap XL the resolution at m/z 400 was 60,000, automatic gain control $5 \times 10^5$ ions, max. fill time, 500 ms. For the Velos Orbitrap Elite the resolution at m/z 400 was 120,000 automatic gain control $1 \times 10^6$ and the maximum ion time 200 ms. A TOP5 (LTQ Orbitrap XL with automatic gain control 10,000 ions, max. fill time 100 ms) or a TOP 25 (Velos Orbitrap Elite with automatic gain control 5000, max. fill time 150 ms) method was applied for collision-induced dissociation of multiply charged peptide ions. On both instruments the normalized collision energy was 35% and the activation Q 0.250. Dynamic exclusion was set to 45 s.

## MS Data analysis
Raw files were searched with MaxQuant version 2.4.9.0[75,76] against the *homo sapiens* Uniprot reference proteome (ID: UP000005640; 20594 protein entries; October 2022).

Essentially, default settings were used in MaxQuant. Trypsin/P was used as proteolytic enzyme and up to two missed cleavages were allowed. A 1% false discovery rate was applied to both peptide and protein lists. Methionine oxidation and N-terminal acetylation were set as variable and carbamidomethylation as fixed modifications. The minimum number of unique peptides was set to 1. Label-free quantification[77] was enabled, with a minimum ratio count of two and the option 'require MS/MS for label-free quantification (LFQ) comparisons' enabled.

For data analysis, the proteingroups.txt file of Maxquant was used and loaded into Perseus 2.0.10.0[78]. Entries for reverse and contaminant hits as well as proteins only identified by site were removed from the analysis. LFQ intensities were $\log_{10}$-transformed. Only protein groups with at least six reported LFQ intensities were considered for further analysis. Missing values were imputed from normal distribution using the following settings in Perseus: width 0.5 and down shift 1.7. A one-sided two sample t-test was performed with a valid value

(non-imputed) filter set for two out of three in the NRBP1/2 IP replicates. Proteins with a *p* value below 0.05 and a student's t-test difference of at least 5 were considered as significantly enriched candidates.

The mass spectrometry proteomics data have been deposited to the ProteomeXchange Consortium (http://proteomecentral.proteomexchange.org) via the PRIDE partner repository[79] with the dataset identifier PXD051452.

## Co-immunoprecipitation (Co-IP)
For the Co-IP experiments, cells were transfected with the respective plasmids and collected 48 h after transfection. The cells were washed twice with PBS and lysed in the CHAPS-based buffer as stated above, supplemented with Complete Protease Inhibitor Cocktail, Phosphatase Inhibitor Cocktail 2, and Cocktail 3. For RNA-dependent Co-IP performed in Fig. 1d,e, cell lysate was additionally treated with either 50 U/mL RNasin (Promega, N2615) to protect RNA, or 40 μg/mL RNase A (Thermo Scientific, EN0531) to digest RNA. The cell lysates were pre-cleared with Dynabeads™ Protein G for 30 min and then incubated with specific antibodies coupled with the Protein G magnetic beads for 2-3 h. After washing the beads with CHAPS-based buffer, the proteins were eluted by boiling at 95 °C for 5 min in Laemmli Sample Buffer (containing 10% glycerol, 1% beta-mercaptoethanol, 1.7% SDS, 62.5 mM Tris pH 6.8, and bromophenol blue).

## Cell lysis and western blot
Cells were washed twice with PBS and then lysed with RIPA buffer (150 mM NaCl, 50 mM Tris pH 8.0, 1% NP-40, 0.5% sodium deoxycholate and 0.1 % SDS), supplemented with Complete Protease Inhibitor Cocktail, Phosphatase Inhibitor Cocktail 2 and Cocktail 3 for 10 min before centrifugation. The protein concentration was measured using Bio-Rad Protein Assay Dye Reagent Concentrate (#5000006) and adjusted to the same level. The cell lysates were mixed with Laemmli Sample Buffer and heated for 5 min at 95 °C. Cell lysates were loaded onto sodium dodecyl-sulfate polyacrylamide gel electrophoresis (SDS-PAGE) gels and the proteins were transferred to polyvinylidene difluoride (PVDF) membranes. The membranes were blocked with 5% bovine serum albumin (BSA) diluted in Tris-Buffered Saline with Tween 20 (TBST) buffer for 1 h at room temperature and then incubated with primary antibodies overnight at 4 °C. Membranes were washed with TBST buffer and incubated with the corresponding horseradish peroxidase (HRP)-conjugated secondary antibodies for 1-2 h. Pierce ECL Western Blotting Substrate (32209) or SuperSignal West Femto Maximum Sensitivity Substrate (34095) were used to detect protein signals. The chemiluminescence signal was captured using a LAS-4000 camera system.

## siRNA knockdown
ON-TARGET plus SMARTpool siRNAs directed against NRBP1(L-005356-00), NRBP2 (L-005340-02), G3BP2 (L-015329-01), ELOB (L-012376-00), ELOC (L-010541-00) and Non-targeting control (siControl) siRNA (D-001810-10) were purchased from Dharmacon. HeLa cells were transfected with a final concentration of either 20 nM siRNA (for NRBP1 and NRBP2 knockdown) or 10 nM siRNA (for ELOB and ELOC knockdown) in two consecutive days using the Lipofectamine RNAiMAX Transfection Reagent (Thermo Fisher Scientific) according to the manufacturer's instructions. For NRBP1 and NRBP2 knockdown, cells were collected for analysis two days after the second siRNA transfection. For Supplementary Fig. 9d, e, cells were transfected with plasmids one day after the last siRNA transfection. Cells were collected 28 h after the plasmid transfection. For G3BP2 knockdown in Supplementary Fig. 3d, trypsinized cells were transfected with 20 nM siRNA using Lipofectamine RNAiMAX. Cells were transfected again with L1-Flag-neo and NRBP2-expressing plasmids using Lipofectamine 2000 (Invitrogen, 11668027) the next day and stained two days later.

## shRNA knockdown

The pTRIPZ doxycycline (DOX)-inducible shRNA constructs targeting NRBP1 and NRBP2 or the non-targeting control sequence (shControl) were obtained from Dharmacon. Information of the target sequence is given in Supplementary Data 3. Viral particles were produced using the Trans-Lentiviral shRNA Packaging mix (Horizon Discovery) according to the manufacturer's protocol. HeLa and MCF-7 cells were transduced with the viral particles in the presence of 8 μg/mL polybrene. Puromycin (2 μg/mL) selection was carried out 48 h post-transduction for one week. shRNA expression was induced with 2 μg/mL DOX for the specified days unless mentioned otherwise.

## Generation of G3BP1 knockout cell lines

To generate G3BP1 CRISPR/Cas9 knockout HeLa cell lines, three target sequences were selected from the human CRISPR knockout pooled library (Brunello, Addgene #73178) and cloned into pSpCas9(BB)−2A-Puro (PX459) V2.0 vector (Addgene # 62988). HeLa cells were transfected with a pool of the three plasmids using Lipofectamine 2000. The pSpCas9(BB)−2A-Puro (PX459) V2.0 vector was also transfected to generate sgControl cells. Twenty-four hours after transfection, cells were selected with puromycin (2 μg/mL) for two days. Monoclonal cell populations were obtained by limiting dilutions. Knockout efficiency was confirmed by Western blot. Target sequences are shown in Supplementary Data 3.

## L1 retrotransposition assay

To test the effects of NRBP1/2, their respective mutants and HPO-11 overexpression on L1 retrotransposition, HeLa cells were transfected with either L1-neo or pcDNA3.1 together with expression plasmids for NRBP1/2, their mutants or HPO-11. For experiments performed in knockdown cell lines, the inducible cell lines were first treated with 2 μg/mL DOX for three days before transfection with the L1-neo or pcDNA3.1 plasmid. DOX was removed until the cells were trypsinized for selection with G418. For experiments carried out in knockout cells, the cells were transfected with only L1-neo or pcDNA3.1. For all experiments, cells were trypsinized 48 h post-transfection, and equal numbers were seeded into 6-well plates and selected with G418 (800 μg/mL) (Thermo Fisher Scientific, 11811031) for 8-14 days. The G418-containing medium was changed every two or three days. To stain the colonies, cells were washed twice with PBS, fixed with methanol for 10 min, and stained with 0.5% crystal violet for 10 min[80]. The number of G418 resistant colonies was counted using ImageJ software. The relative retrotransposition activity was calculated by dividing the colony numbers obtained from L1-neo-transfected cells by those from pcDNA3.1-transfected control cells, thereby controlling for potential toxic effects caused by overexpression or knockdown.

## Cell viability assay

The MTT assay was performed to evaluate the potential cytotoxic effects of NRBP1 knockdown and NRBP2 overexpression. For NRBP1 knockdown, HeLa cells stably expressing NRBP1 shRNAs were induced with 2 μg/mL DOX for 5 days. For NRBP2 overexpression, HeLa cells in a 24-well plate were transfected using Lipofectamine 2000 with the same plasmids as those used in the L1 retrotransposition assay conducted in a 6-well plate. To accommodate the smaller well size, plasmid amounts and reagent volumes were proportionally scaled down. The MTT assay was conducted according to the manufacturer's protocol (Roche, 11465007001). Briefly, the cells were incubated with MTT labeling reagent for 4 h at 37 °C, followed by solubilization of the formazan crystals using the provided solubilization buffer overnight at 37 °C. Absorbance was measured at 590 nm using a microplate reader. A standard curve was generated using known cell numbers to ensure the accuracy and validity of the data. All experiments were performed with three independent biological replicates, and each biological replicate included at least three technical replicates.

## RNA immunoprecipitation (RIP)

For endogenous ORF1p RIP, MCF-7 shControl and shNRBP1-1 cells were treated with 2 μg/mL DOX for 3 days, followed by RIP using an anti-ORF1p antibody. For RIP in NRBP1/2 overexpression conditions, HeLa cells were co-transfected with L1-Flag-neo and either HA-NRBP1, HA-NRBP2, or empty vector for 48 h, and subjected to RIP using an anti-Flag antibody. For RIP in NRBP2 knockdown cells, HeLa shControl and shNRBP2-1 cells were treated with 2 μg/mL DOX for 3 days, then transfected with the L1-Flag-neo reporter plasmid for 48 h, followed by RIP using an anti-Flag antibody.

Cells were washed twice with cold DPBS and then lysed in NP-40 Buffer containing 50 mM Tris pH 8.0, 150 mM NaCl, 1% NP-40, protease inhibitor and RNase inhibitor for 30 min. After centrifugation at 13,000 g for 20 min at 4 °C, 10% of the cleared lysate was reserved as input for total cellular L1 mRNA quantification and normalization of RIP signals, and an additional 10% was set aside for Western blot analysis. The remaining lysate was pre-cleared by incubating with Protein G or Protein A magnetic beads (depending on the antibody used) for 30 min at 4 °C. The pre-cleared supernatant was then incubated with ORF1p or Flag antibody overnight at 4 °C. The beads were washed three times. Approximately 10% of the beads was retained for Western blot analysis to validate the efficiency of the immunoprecipitation. The remaining beads were treated with DNase I for 30 min at 37 °C. The reaction was stopped by EDTA. RNA was extracted using TRIzol (Invitrogen, 15596018). Details of qRT-PCR analysis are provided in the following section.

To quantify the overall effect of NRBP1/2 on L1 mRNA levels, different normalization strategies were applied. For HeLa cells transfected with L1 plasmids, quantification was normalized to the Hygromycin resistance gene, which is encoded by the same plasmid. This normalization approach eliminates the influence of differences in transfection efficiency[81]. For MCF-7 cells, which express endogenous L1, GAPDH was used as an internal control.

Quantification of the co-precipitated L1 mRNA was carried out using the Imprint®RNA Immunoprecipitation (RIP) Kit from Sigma with some modifications. To account for differences in input RNA amounts, the relative enrichment of L1 mRNA in RIP samples was calculated using the following formulas:

$$\Delta C_t(\text{normalized RIP}) = C_t(\text{RIP}) - [C_t(\text{Input}) - \log_2(\text{Input dilution factor})]$$

$$\Delta\Delta C_t = \Delta C_t(\text{Experimental group}) - \Delta C_t(\text{Control group})$$

$$\text{RIP fold enrichment} = 2^{-\Delta\Delta C_t}$$

To control for differences in ORF1p immunoprecipitation efficiency between control and experimental samples (e.g., due to knockdown or overexpression), the RIP fold enrichment values were further adjusted based on the ratio of ORF1p protein levels in the IP fractions of the two groups. The final data were normalized to the control group, which was set to 1. Overall, L1 mRNA enrichment was normalized to both input RNA levels and precipitated ORF1p protein levels.

## Quantitative reverse transcription PCR (qRT-PCR)

Total RNA extraction and DNA digestion were performed using FastGene RNA Premium Kit (Nippon Genetics, FG-81050) unless otherwise specified. cDNA was synthesized with Oligo-dT primer by using Transcriptor High Fidelity cDNA Synthesis Kit (Roche, 5081963001).

qRT-PCR was performed using Luna Universal qPCR Master Mix (NEB, M3003) running on the Light Cycler 96 System (Roche). Control reactions without reverse transcriptase (no RT) were performed to confirm the absence of contaminating DNA. The data were analyzed by the relative quantification $2^{-\Delta\Delta Ct}$ method[82]. GAPDH was used as the reference gene for normalization, unless otherwise specified. All qRT-PCR primers used in this study are provided in Supplementary Data 3.

## RNA sequencing (RNA-seq)

The RNA-seq data were analyzed on the European Galaxy server (usegalaxy.eu). The quality of FASTQ files was checked by FastQC (v0.72). Cutadapt (v1.16.5) was used to remove the adapters[83]. Trimmed reads were aligned to the human genome USCS build hg38 using RNA STAR (v2.7.2b)[84]. The number of reads was counted by FeatureCounts (v1.6.4) using default parameters[85]. Differentially expressed genes were identified by EdgeR (v3.24.1)[86]. Genes were considered to be significantly differentially expressed when the FDR < 0.05, with $\log_2$ fold change ≥0.58 (1.5-fold change) for upregulated genes and ≤ −0.58 (1.5-fold change) for downregulated genes. The RNA-seq data are available in Supplementary Data 2. RNA-seq data have been deposited in BioProject under ID PRJNA1101872.

## Immunofluorescence and confocal microscopy

Cells seeded on coverslips were washed with PBS, fixed in 4% PFA for 20 min, and permeabilized with 0.1% Triton-X100 / PBS for 5 min. The cells were then incubated in a blocking solution (3% FBS in PBS) for 20 min and with the indicated primary antibodies for 1 h at room temperature. The following primary antibodies were used: Rabbit anti-ORF1p (1:25 dilution, Abcam, ab230966), Mouse anti-DDX6 (1:100 dilution, Sigma, SAB4200837), Mouse anti-Flag (1:400 dilution, Sigma, F1804), Mouse anti-G3BP1 (1:200 dilution, Santa Cruz, sc-81940), Rabbit anti-G3BP2 (1:200 dilution, Proteintech, 16276-1-AP), Rat anti-HA (1:200 dilution, Roche, 11867423001), Mouse anti-TIAR (1:50 dilution, Santa Cruz, sc-398372), Rabbit anti-Myc (1:200 dilution, Cell signaling, 2278) and Rabbit anti-Flag (1:400 dilution, Cell signaling, 14793). Subsequently, the cells were incubated with the corresponding Alexa Fluor-coupled secondary antibodies for 40 min at room temperature, and the nuclear DNA was stained with Hoechst 33342 (Sigma, H3570). ProLong Gold antifade (Invitrogen, P36930) was mounted on top of the cells. Confocal microscopy was performed using either an LSM-U-NLO or LSM-I-NLO confocal microscope (Carl Zeiss).

## Single-molecule RNA fluorescence in situ hybridization (smFISH)

MCF-7 shControl and shNRBP1-1 cells were treated with 2 µg/mL DOX for 3 days prior to smFISH analysis. In total, 48 Stellaris RNA FISH probes, each 20 nucleotides in length targeting L1 mRNA, were produced and purified by Biocat[87]. The probes were conjugated with Quasar 670 (sequences of the probe sets are available in Supplementary Data 3). ORF1p was stained with the ORF1p antibody (Abcam, ab230966) at a 1:25 dilution. The smFISH staining was performed according to a published Stellaris RNA FISH and IF co-staining protocol (https://www.biosearchtech.com/support/resources/stellaris-rna-fish). Images were captured using a confocal LSM-I-NLO microscope (Carl Zeiss) with Airyscan super-resolution mode. The probe sequences for *fat-7* and *trcs-1* are provided in Supplementary Data 3.

## Rough endoplasmic reticulum (ER) fractionation

Rough ER fractions were isolated from HeLa cells according to the Endoplasmic Reticulum Enrichment Kit protocol (Novus Biologicals, NBP2-29482). Full-length ATF6 (1:1000 dilution, Santa Cruz, sc-166659), NRBP1/2 (1:1000 dilution, Proteintech, 21549-1-AP), and GAPDH (1:10000 dilution, Proteintech, 60004-1-Ig) were detected by Western blot.

## 3TC and L1 siRNA treatments

HeLa cells were transfected with 20 nM siControl or siNRBP2 for two consecutive days (day 1 and day 2), with or without 10 µM 3TC (Sigma, L1295) added to the medium. On day 3, the medium was replaced and 10 µM 3TC was either added to the treatment group or omitted in the control group. Cells were collected for RNA extraction 24 h later.

For the knockdown of endogenous L1 mRNA, an siRNA was designed based on a naturally occurring L1-targeting siRNA and synthesized by Sigma. The target sequence is 5′-CCCAGGCTTGCTTAGG-TAAACA-3′[88]. 20 nM siL1 was transfected either alone or with 20 nM siNRBP2 for two consecutive days. RNA was extracted two days after the last transfection.

## Cycloheximide (CHX) chase assay to examine NRBP1 protein half-life

HeLa cells were transfected with either HA-NRBP2 or HA-NRBP2-dNRB expression plasmids using Lipofectamine 2000. After 48 h, cells were treated with either DMSO or 40 µg/mL CHX (Sigma, 239765) and collected at the indicated time points for Western blot. Protein levels of NRBP1 (1: 1000 dilution, GeneTex, GTX84003), HA (1: 1000 dilution, Roche, 11867423001) and GAPDH (1: 10000 dilution, Proteintech, 60004-1-Ig) were detected. For quantification, NRBP1 protein levels were normalized to GAPDH. In each treatment group, the protein level at 0 h post-CHX treatment was set as 1 for normalization.

## Ubiquitination assay

HEK293T cells were transfected with plasmids expressing 6His-Ub, NRBP1-Flag, and NRBP2-Myc using PEI (Polysciences, 24765-100). After 48 h, the cells were treated with 20 µM MG132 for 2.5 h before collection and then trypsinized. One-tenth of the cells was lysed in RIPA buffer and saved as input. The remaining cells were sonicated in Lysis Buffer (50 mM HEPES (pH 7.6), 6 M GuHCl, 5 mM imidazole, 10% glycerol, 0.2% NP-40) followed by centrifugation to collect the supernatant. The supernatant was then incubated overnight at 4 °C with TALON Metal Affinity Resin (Takara, 635501) pre-washed with Wash Buffer (50 mM HEPES (pH 7.6), 300 mM NaCl, 7.5 mM imidazole). The next day, the resin was washed twice with a 1:3 mixture of Lysis Buffer and Wash Buffer, followed by two washes with Wash Buffer. Bound proteins were eluted by adding 1× Laemmli Sample Buffer and heating at 95 °C for 10 min.

## Proteasome, calpain and lysosome inhibitors treatments

HeLa cells were transfected with either an empty vector or an HA-NRBP2 expression plasmid. Thirty-two hours after transfection, the cells were treated for 16 h with either DMSO or one of the following inhibitors: 10 µM MG132 (Sigma, 474790), 100 nM BafA1 (Cell signaling, 54645), 0.1 µM or 1 µM PS-341 (Sigma, 5.04314), 0.1 µM or 0.2 µM Epoxomicin (MedChemExpress, HY-13821), 20 µM or 50 µM Calpeptin (Sigma, 03-34-0051). Cell lysates were collected in RIPA buffer for Western blot.

## Phylogenetic analysis of *NRBP*

To identify homologs of human *NRBP1* and *NRBP2*, we performed BlastP (vs. 2.15.0 + ) searches for both proteins in UniProtKB/Swiss-Prot and UniProtKB/TrEMBL (release 2024_01) using an E-value cutoff of 1e-28[89–91]. This threshold was required to prevent the detection of unrelated protein kinase families. Subsequently, we tested the resulting 8,345 sequences with a reverse BlastP search in the human proteome (ID: UP000005640). If either NRBP1 or NRBP2 was the top-ranking hit, we kept the respective sequence, otherwise it was rejected. After quality filtering for 80% or higher NRBP1/2 coverage in the BlastP alignments and a further filtering step for the representation of the complete predicted folded region of NRBP1/2 from AlphaFold[58,59] with a tolerance of 40 amino acid residues, we obtained a set of 2065 proteins. We obtained the taxonomic information of the respective species using a service from The European Nucleotide Archive (ENA)[92].

In order to conduct phylogenetic analysis, both data sets were merged and sequences were aligned using MAFFT v.7.487 with 1000 iterative refinements[93]. Alignment masking was performed with trimAl v.1.4.1 by removing all alignment positions containing gaps in 10% or more of the sequences[94]. Phylogenetic inference was computed with the maximum likelihood method using IQ-TREE v.2.1.2 in conjunction with model selection via ModelFinder to identify the most fitting model of sequence evolution as well as utilizing 1000 ultrafast bootstraps to assess branch support[95–97]. Tree evaluation and graphics were done with iTOL v6.9[98]. Leaf distances were measured with the Bio.Phylo package using clade labels exported from iTOL[99].

### Sequence analysis of the pseudokinase domains of NRBP

We predicted the absence of protein kinase activity by scoring three amino acid residues of the catalytic triad, comprised of the ATP-binding β3-lysine, the catalytic aspartate within the catalytic loop HRDXXXN motif, and the metal binding aspartate of the activation loop DFG motif[67]. If two of the three amino acid residues did not match the protein kinase consensus, we rated the proteins as pseudokinases. For the identification of the respective amino acid residues, we generated HMMSEARCH (vs. 3.3, http://hmmer.org/) alignments with the PFAM protein kinase domain PF00069 for each sequence[70].

### Statistics and reproducibility

Experiments in this study were independently repeated two to seven times. Most of the experiments were performed three times. For those repeated twice, results were highly reproducible. Experiments with greater variability (e.g., Fig. 1k) were repeated more times due to higher data variation. The number of biological replicates is indicated in each figure legend. Pearson's correlation coefficients (r) were calculated using the PEARSON function in Microsoft Excel.

### Reporting summary

Further information on research design is available in the Nature Portfolio Reporting Summary linked to this article.

## Data availability

The mass spectrometry proteomics data generated in this study have been deposited in the ProteomeXchange Consortium via the PRIDE partner repository under accession code PXD051452. The RNA-seq data have been deposited in BioProject under ID PRJNA1101872. Uncropped Western blots are available in Source Data. Source data are provided with this paper.

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

## Acknowledgements

We thank the ProteomeXchange Consortium for providing globally coordinated standard data submission. We thank the staff of Life Imaging Center (LIC) at the University of Freiburg for providing access to their microscopy resources and for their outstanding technical support. We thank Emmanuelle Moraes Ribeiro, Wolfgang Schamel, Marie Jöst, and Sonja-Verena Albers (all from the University of Freiburg) for kindly providing the proteasome inhibitors. We thank Karin Forsberg Nilsson (Uppsala University) for valuable discussions. We thank the China Scholarship Council (CSC) for providing financial support to the work of S.C. and R.L. in the lab of R.B. B.W. was supported by the Deutsche Forschungsgemeinschaft (DFG, German Research Foundation), transregional collaborative research grant TRR130, SPP 2453 (Project-ID 541758684; WA 1598/7-1), and GRK 2243/2 (Project-ID 285767414). This work was funded by the Deutsche Forschungsgemeinschaft (DFG; German Research Foundation SFB 1381, Project-ID 403222702) and Germany's Excellence Strategy (CIBSS-EXC-2189, Project-ID 390939984) to R.B. This project was further supported by DFG funding to W.Q. (GZ: QI 118/4-1, Project-ID 537144839).

## Author contributions

W.Y., W.Q. and R.B. conceived/designed the experiments, analysed data, and wrote the manuscript. W.Y. and S.C. performed most of the experiments. R.L. conducted and analyzed the qRT-PCR shown in Fig. 6e and Supplementary Fig. 10b. R.G. contributed to the rough ER extraction experiment. X.C. assisted in preparing slides for confocal microscopy. S.U. performed the experiments in Fig. 4e and Supplementary Fig. 7e, i. D.O. did IP in Supplementary Fig. 1c. P.E. and A. Trentino contributed to cloning. J.S., K.F., A. Thien, and B.W. contributed to mass spectrometry experiments and data analysis. T.H. provided material support. T.S. and E.S. performed phylogenetic analyses and contributed to writing the respective sections of the manuscript.

## Funding

## Competing interests

The authors declare no competing interests.
