## [Transparent Peer Review file · Nature Communications]

Opposing roles of pseudokinases NRBP1 and NRBP2 in regulating L1 retrotransposition

Corresponding Author: Dr Wenjing Qi

Version 0:

Reviewer comments:

Reviewer #1

(Remarks to the Author)

This paper by Yang and colleagues explores the NRBP1/2 paralog proteins and their opposing regulation of LINE-1 retrotransposons with effects on L1 RNA binding within its RNP. Evidence is presented that these opposing effects are not due to competition between the two paralogs but instead due to NRBP2-mediated targeting of NRBP1 for degradation. Many proteins capable of suppressing L1s in cell culture have previously been identified, but few have been shown to increase retrotransposition, and the reported interplay of two closely related proteins in L1 activity is novel and of general interest. However, additional controls, quantifications, and clarifications of some statements are needed to strengthen conclusions. In my opinion, this precludes publication of the paper in its present version.

Notably, although an important conclusion of the paper is that NRBP1 is targeted for proteosomal decay by NRBP2, this is based largely on use of the proteasome inhibitor MG132 (which may also have off-target effects). Additional supporting data would strengthen this conclusion. For example, is ubiquitination of NRBP1/2 detected in MS data? Can NRBP1 be detected/IPed by anti-Ub antibodies? Does subcellular localization track with any proteasome marker antibodies?

Also, enthusiasm for the paper is reduced somewhat because the authors failed to determine a mechanism for NRBP1 degradation.

In addition, the conclusion that NRBP2 regulation activates inflammatory pathways and that this may be caused by L1 is suggestive but rather weak, being based mainly on RNA-Seq and GO analyses and comparison between NRBP2 KD stimulation of mRNAs and those stimulated by ORF1p (as reported by another group in different cells).

Cell toxicity/proliferation analyses for NRBP1/2 are required to confirm no effects of their overexpression or knock-down. This is important for both the retrotransposition studies and to show that altered expression of these proteins themselves do not stress cells (a possibility the authors' consider in the Discussion, p. 22). For example, the loss of overall expression of about 30 percent evident in Fig. 3b could indicate some toxicity of NRBP1 KD.

p. 7, lines 143; p. 51, line 1169; p. 8, line 165, etc.. It cannot be stated that NRBP1/2 interacts with ORF1p because no IP in the presence of RNase was tested as a control. This needs to be made clear in the text - or these controls need to be performed. All of these proteins are RNA-binding proteins and co-IP may be the result of RNA tethering, not direct protein-protein binding. The statement "NRBP1 and NRBP2 might be novel components of the previously described L1 RNP complexes" is acceptable.

p. 8, line 170. The statement "The number of ORF1 containing puncta increased upon knockdown of NRBP1 (Fig. 2a)" requires some kind of quantitation.

The statement, "As the ORF1 containing puncta upon NRBP1 knockdown did not show an enrichment of the SG marker G3BP1 (Fig. 2a), they are probably not SGs", may be true but requires more elaboration in light of previous publications. A couple of papers have noted that ORF1p can also enter P-bodies in unstressed cells and this should be noted (in stressed cells P-bodies and SGs tend to coalesce). Furthermore, do other SG markers such as TIA-1 also not colocalize? Of possible relevance, in Goodier et al (PLOS Genetics 2022) tagged ORF1p could still form cytoplasmic granules in G3BP1/2 double-knockout U2OS cells, although it appears in Fig. 5H they did not colocalize with TIA-1.

p.9, line 179, Fig. 2b. "In control cells, we observed a partial colocalization of ORF1 and L1 mRNA in cytoplasm, while NRBP1 knockdown led to a strongly decreased colocalization of the protein and mRNA (Fig. 2b)." After staring at these micrographs, the change in colocalization of ORF1 and L1 RNA is not convincing for me - perhaps quantification would help.

p. 10, line 210, Ext Fig. 2b. It should be noted here that G3BP2 can partially compensate for G3BP1. Double-knockouts were not attempted here.

Also of more minor significance:

p. 8, line 151, Fig. 1e-h. I would suggest that the rationale for using pcDNA3 as a transfection control (has neoR) should also be noted in the text or figure legend (not just in the Methods).

p. 8, line 170, Fig. 2a top. Mark the "few small punctate structures" with arrows.

p. 8, line 171-2. I believe the statement "Sequestration of ORF1 and L1 mRNA into stress granules (SGs) was shown before to inhibit L1 retrotransposition" could be tempered. Manipulation of SG proteins for effects on L1 by this previous SAMHD1 study, in my opinion, did not definitively prove SG sequestration was the direct cause - SG changes may have been correlative. Same for p. 10, line 200, "... G3BP1 knockdown has been shown to lead to the loss of large ORF1 foci, which in turn activates L1 retrotransposition⁴⁰" - not adequately proven in my opinion.

p. 11, line 239, Fig. 3g. The perinuclear reticular staining pattern is not obvious to me. Arrows might help. Curiously, in this particular figure, cotransfected cells seem to have greater NRBP1 signal than cells transfected with NRBP1 alone, which is inconsistent with the paper's conclusion that NRBP2 targets NRBP1 for degradation?

p. 13, line 278. "We next asked whether loss of inhibition in NRBP2-N is caused by its loss of ORF1 binding. We found both NRBP2-N and NRBP2-C could be co-immunoprecipitated with ORF1, implicating that NRBP2 might interact with ORF1 through multiple interfaces".

Again, this statement needs to be modified because it cannot be concluded that NRBP2 binds ORF1p because no RNase controls were run. Furthermore, an alternative explanation is that NRBP2 has two binding sites that may interact with RNAs that bind ORF1p or proteins that bind RNAs that bind ORF1p.

p. 14, line 310, Ext Fig. 7a. NRBP1 band intensities should be quantitated as a function of GAPDH bands (which also appear to vary somewhat in intensities)

line 289. The meaning of the term "half-switches" is unclear.

Reviewer #2

(Remarks to the Author)

Long Interspersed Element 1 (LINE-1 or L1) is the only autonomously active retrotransposon in modern humans. L1 retrotransposition poses a mutagenic threat to genome integrity, and numerous host defence mechanisms have evolved to restrict L1 retrotransposition, acting at various stages of the retrotransposition process. In this manuscript, Yang et al. identify two gene paralogs, NRBP1 and NRBP2, as regulators of LINE-1 retrotransposition. Intriguingly, these proteins have opposite impacts upon retrotransposition. The authors claim that NRBP1 enhances retrotransposition by facilitating the association between L1 mRNA and ORF1p in the L1 ribonucleoprotein particle, and that NRBP2 diminishes retrotransposition by targeting of NRBP1 for proteasome-mediated decay by NRBP2 likely via heterodimerization.

The opposite impacts of NRBP1 and NRBP2 on L1 retrotransposition are well demonstrated. However, the experimental evidence that NRBP1 enhances retrotransposition efficiency by facilitating the interaction between ORF1p and L1 mRNA is the weakest part of the paper. There is much stronger experimental support for NRBP2-mediated proteasomal degradation of NRBP1. While many experiments are well performed, there are numerous important controls missing throughout the paper, which renders the results difficult to interpret.

Major Points:

1. The methods section needs additional detail across most experiments. For example, what was the concentration of antibodies used in immunofluorescence experiments? How did the author carry out the inhibition with MG132 in Fig.5? What L1 sequence was used to generate the smFISH probes?

2. The retrotransposition assays in the presence of NRBP1/2 knockdown and overexpression in Figure 1 are very nicely performed and presented. The inclusion of individual dots on the bar graphs for each replicate is greatly appreciated. However, it is unclear from the figure legend and methods what constitutes a biological replicate. Standard practice is to average three wells (technical replicates) to arrive at one biological replicate, and perform 3 or more independent biological replicates on different days (PMID: 26895052). Could the authors please clarify what is indicated by a biological replicate in this publication? Also, it is mentioned that seven biological replicates were performed for Figure 1J, while only three were performed in figures 1F, 1H, and 1L. Were the additional replicates done to achieve statistical significance between NRBP2 knockdown and control? A bit of transparency around this choice would be appreciated.

3. Many cellular factors that influence retrotransposition efficiency interact with the L1 RNP via an RNA-dependent interaction. Replication of the co-IP experiments in Figure 1, +/- RNaseA, would provide more insight regarding the potential mechanism of retrotransposition modulation by NRB1/2.

4. The impact of NRB1 kd and NRB2 overexpression on the association of ORF1p with L1 mRNA shown in Figure 2C and 2F is, at first glance, striking, but it is unclear how the experiment was performed and analysed. Generally, the amount of L1 mRNA associated with ORF1p should be shown as enrichment over the amount of L1 mRNA present in the input (before IP), with GAPDH serving as an internal control. It does not appear that L1 mRNA levels in the input were quantified at all, as part of this experiment or to determine whether NRB1/2 impact L1 mRNA expression levels in general. Please replicate the RNA-IP experiment and show the fold enrichment in the IP fraction relative to input. Please also consider showing RT-PCR quantification of L1 mRNA levels upon NRB1/2 knockdown and overexpression—this suggestion is also highly relevant to point 11, below.

5. For colocalisation experiments, ORF1p was detected using an anti-ORF1p antibody, and L1 RNA was detected using FISH probes across the entire L1 mRNA sequence. This approach likely detects the majority of ORF1p and L1 mRNA molecules in the cell, including those arising from expression of defunct, older subfamily L1 copies that may not be associated in active RNPs. Previous publications including PMID: 20949108 have used a tagging strategy to detect L1 mRNA and both ORF1p and ORF2p arising from the same, active, ectopically expressed L1 element. As the L1 RNA and proteins exhibit a strong cis preference—i.e., ORF1p and ORF2p preferentially bind to their encoding mRNA molecule—this approach has a tremendous advantage in allowing detection of ORF1p, ORF2p, and L1 mRNA that are incorporated into potentially active RNPs without the background noise of endogenous, potentially inactive L1 products. In Figure 2B, the partial colocalization of L1 mRNA and ORF1p, and its dissolution in NRB1 knockdown cells, is not very convincing. The use of tagged expression constructs would provide a much stronger, complementary experiment to demonstrate that L1 ORF1p and mRNA interaction is interrupted when NRB1 expression is reduced. It would also allow determination of the subcellular localization of ORF2p, the endogenous expression of which is nearly impossible to detect, and potential impacts of NRB1 knockdown on its cellular distribution and association with ORF1p and L1 mRNA.

6. Please provide control images for the immunofluorescence and smFISH experiments and include the description of those experiments in the methods.

- a. A negative control experiment for the smFISH using probes targeting RNA not present in the cells (i.e. bacterial gene) would greatly improve the confidence in the FISH results. A control experiment carried out on DNase treated cells would also provide confidence that the probes are not off target binding any DNA.
- b. Negative controls for the immunofluorescence would ideally include an IgG control to confirm there is no off-target binding.

7. As the same antibody is used to detect NRB1 and NRB2, it would be nice to see both proteins on the gel fragment shown for experiments in which one or the other is knocked down (e.g., Figure 1F, H, J and I), to confirm that shRNA against one does not impact expression of the other. If the NRB1 and NRB2 nucleotide sequences are dissimilar enough that qRT-PCR primers can be designed to distinguish their mRNAs, this would be another possible approach to confirm the specificity of knockdown for each paralog.

8. The inclusion of knockdown and overexpression of both NRB1 and NRB2 in Figure 1 is excellent and a point of strength for the paper. However, this reciprocity unfortunately is not continued in Figure 2. What is the impact of NRB1 overexpression on ORF1p localization? Upon RIP of L1 RNA with ORF1p? Similarly, what is the impact of NRB2 knockdown on ORF1p localization and RIP of L1 RNA with ORF1p? Finally, do NRB1 and ORF1p show colocalization by IF?

9. In Figure 2, it is difficult to have confidence in the proposed co-localisation of ORF1p and the L1 mRNA from these images. In particular the shControl where authors have pointed out regions of co-localisation with yellow arrows appear to be a result of ubiquitous cytoplasmic distribution or possibly even background fluorescence. Negative control images here will greatly improve confidence in this result. In Figure 2E, it is very clear that ORF1p-FLAG and G3BP1 co-localise to large granules, but it looks like HA-NRB2 is uniformly distributed. This image does not definitively show co-localisation between these three proteins.

10. In Figure 3G and 5B, can the shift in NRB1 subcellular localization in the presence of NRB2 be quantified across multiple cells/images, using image analysis software? The shift is noticeable by eye, but few representative cells are shown and quantitative data would be much more convincing.

11. Does NRB2 knockdown (or NRB1 overexpression) actually increase L1 mRNA? The “input” gels for the co-IP experiments would suggest this is not the case for L1 ORF1 protein expression. A previous publication (PMID: 36639706) demonstrated that at least in HeLa cells, the immune response to L1 overexpression is caused by L1 mRNA, not L1 cDNA. Demonstration that NRB2 knockdown or NRB1 overexpression increases L1 mRNA expression would make the conclusion that immune activation in response to NRB2 knockdown occurs through L1 activation more solid.

12. In Figure 6F, the number of RA patient samples analysed is much higher than the number of healthy controls. Why not analyse equal numbers of patients for each group?

Minor Points:

1. It was a bit difficult to find the identity of the anti-ORF1p antibody used in the materials and methods. It would be helpful to

mention it in the Western blot section which comes before the RNA FISH/IF section where it currently is first mentioned.

2. Normally RNA-seq data would be validated by RT-qPCR for up and down regulated genes in separate biological replicates and the correlation of those data measured. It's unclear whether the qPCR data from Fig.6 e is from separate biological replicates. If the data do represent separate biological replicates, a correlation analysis would strengthen the conclusions from this experiment.

3. What was the rationale behind carrying out some experiments in MCF-7s and some in HeLa cells? The experiments switch between cell types with no explanation.

4. Line 311 MG132 referred to as a protease inhibitor but I believe it's a proteasome inhibitor

Reviewer #3

(Remarks to the Author)

Reviewer #4

(Remarks to the Author)

The authors describe the intricate mechanism of two new cell factors regulating the mobilisation of human L1 retrotransposon. Interestingly, they describe the evolutionary occurrence of paralog of NRBP1, termed NRBP2, in such way that NRBP1 has a positive role in favouring L1 retrotransposition and NRBP2 destabilises NRBP1 by heterodimerisation. Based on their results, NRBP1 enhances L1 retrotransposition by facilitating L1 RNP assembly. These results increase the knowledge in both directions: L1 profiting from cellular factors to accomplish its mobilisation and the occurrence of cell factors controlling L1 activity. Interestingly, in this work the authors have been able to track the evolutionary timing of the occurrence of this mechanism controlling L1 mobilisation and, to a great extent, its mechanism of action.

Major comments concerning the text and experimentation are:

1. The title and the abstract are a bit confusing and may initially mislead the reader to understand the roles of NRBP1 and NRBP2 in the opposite way. I suggest rephrasing the title to something similar to (disclaimer: I am not a native English-speaker, so just take this as a suggestion): 'NRBP1 targeting for degradation by its paralog NRBP2 accounts for their opposite roles in regulating LINE-1 retrotransposition'.

2. The introduction should be reshaped. The authors present NRBP1 and 2 through their oncogenic and tumour-suppression roles and this creates a link with the fact that these proteins may regulate the mutagenic impact L1-retrotransposition, but the only biomedical impact that they conclude in the article is potentially through the retrotransposition-independent L1 role in INF-induction in autoimmunity (which may not necessarily need RNP assembly (Fukuda et al. 2021 PNAS). If possible, some more evidence should be built in the introduction if these proteins have any described links to autoimmunity or inflammation.

3. In the section 'NRBP1 and NRBP2 interact with L1-encoded ORF1', the authors confirm by Co-IP that NRBP1 and NRBP2 are bound to ORF1p but it remains unclear if this binding could be due to direct interaction or through L1 RNA or other L1 RNA-bound factors. Treatment of the protein extracts with RNase A would resolve this doubt and help fully elucidating the mechanism of action. For instance, NRBP1 could help other L1 RNA-bound factors that ultimately enhance L1 RNP assembly.

NRBP1 KD would cause an excess of free ORF1p, which concentrates to SG or SG-like bodies. However, it is unclear if this is simply the fate of free ORF1p in excess or because non-SG ORF1p requires from NRBP1. The use of a stable ORF1p mutant with disabled RNA binding ability (i.e. mutants assayed by Luqman-Fatah et al. Nat Commun. 2022) could help to elucidate if the fate of naturally free ORF1p is the same as in their NRBP1 KD or NRBP2 overexpression scenarios. In this same line, it could be interesting to test whether NRBP1 (as potential interactor with ORF1p) colocalises with ORF1p in the scenarios they have tested, as they do with NRBP2 in Fig.2e. They could use the HA-NRBP1 construct used in Fig. 3g the same way that they use HA-NRBP2 in Fig. 2e. By the mechanism of action they present, NRBP1 could have a more likely direct interaction with ORF1p and, therefore, this result would be as or more interesting than the colocalisation with NRBP2.

Finally, since NRBP1 KD does not seem to reduce the levels of ORF1p (Fig. 2d), it could be assumed that L1 RNA has not been affected either. However, since the amount of L1 RNA can only be inferred by the FISH images, a quantification of total L1 RNA by RT-qPCR could be useful to see if the lack of ORF1p binding to L1 RNA triggers the degradation of the latter.

4. In the section 'NRBP1 and NRBP2 contrarily regulate ORF1 and L1 mRNA association', the authors correctly justify the use of MCF-7 cells to study the localisation of ORF1p (Fig. 2a) because they have higher endogenous ORF1p levels than HeLa. And they also use these cells for the study of ORF1p-L1 mRNA association (Fig. 2b). However, they do not provide an explanation of why they resort to ectopic expression of tagged NRBP2 and tagged L1 in HeLa to test the inhibitory role of NRBP2. A priori, they should/could try to see the cellular localisation and levels of endogenous ORF1p in MCF-7 by transfecting just an NRBP2 expressing plasmid. This leads to the testing of SG-localisation of L1 in G3BP1-KO HeLa cells

instead of MCF-7 (Fig. 2e). Thus, the 'inconsistency' in the kind of granules that L1 moves to in NRBP1 KD than in NRBP2-overexpression cannot be ruled to be due to consequences derived from the targeted NRBP, or simply because the used cellular model. This can be resolved by checking if this discrepancy in the properties of the granules still occurs in NRBP1 KD in HeLa cells and NRBP2 overexpression in MCF-7.

5. The experimentation leading to elucidate the role of NRBP2 on NRBP1 and the way the authors conclude that it is due to heterodimerisation is very elegant and I congratulate the authors for that, as well as for the tests resulting in IFN-pathway activation by NRBP2 overexpression. The use of RT-inhibitors could be explored in order to determine if the IFN-pathway activation is occurring through ectopic cytosolic L1 cDNA as described in other scenarios (i.e. Fukuda et al. 2021 PNAS, De Cecco et al. 2019 Nature). However, the link to rheumatoid arthritis appears weak. The authors have not mentioned this disease in the introduction nor any link with NRBP2 or with L1. It would be interesting to have a nicer explanation of why this disease has been tested instead of any other one previously described to have increased L1 activity. Additionally, this link will be more robust if the authors explore the presence of L1 ectopic cDNA or other species that could justify that the innate immunity component in this disease could be (partially) influenced by accumulation of toxic L1 nucleic acids that could be accounted to NRBP2 downregulation.

6. Finally, the phylogenetic analysis determining the occurrence of NRBP2 together with the occurrence of type II ORF1p in L1s is very clever and the results are fascinating and worth to be explored in the future.

Minor comments are mostly related to the need of some further text proofreading. Some small things I have detected are:

- Line 88: The acronym of LINE should be fully displayed here or in the abstract (depending of journal rules) and a final abbreviation must be chosen for the entire paper (in the abstract, the authors use LINE1 and in the paper they use L1 for most of the text).
- The authors use 'ORF1' and 'ORF1 protein' indistinctly to refer to ORF1 protein. In the L1 field, ORF1p and ORF2p are typically used to facilitate distinguishing when the author is referring to protein instead of the ORF.
- Uniformity of units and other conventions: for instance, the authors use 's' for seconds instead of 'sec', while they use 'min' and 'minutes'; they use both knock-down and knockdown; etc.

Sanchez-Luque FJ, PhD

Reviewer #5

(Remarks to the Author)

In this study, the authors identified that, by substantial biochemical approaches, NRBP1/2 as a novel LINE regulator in MCF7 and HeLa cells. Interestingly, the authors also found that NRBP2 evolutionarily acquired later than another paralog NRBP1 stimulates NRBP1 proteasomal degradation. Then the authors narrowed down the functional detail based on these phenomena, hence proving different paralogs involved in human pathogenesis.

The immunohistochemistry, proteomics, RIP-qPCR, and co-IP analyses nicely showed that NRBP1 and NRBP2 regulate differently in LINE1 retrotransposition; however, the relevance between human RA pathogenesis and the two paralogs' inverse correlation is rather preliminary and the data are not conclusive.

Overall, I believe that before being considered for publishing, a significant change would be necessary.

<Major>

Q1) What is the experiment's scientific rationale for using MCF7 and HeLa for all biochemical assays? I only noticed the amount of ORF1 in both cells in Extended Figure 2a, but I'm curious to know if NRBP1 and NRBP2 are expressed in all tumor cells or not at least. I would also like to know the experimental justification and explanation for your selection of these two cells.

Q2) Is there a rationale behind not purifying each NRBP1 and NRBP2 complex separately?

Q3) Even though NRBP2 overexpression is more prominent to prove the inhibitory role of NRBP2 on L1 than knockdown, the authors should confirm the several studies using shRNA of NRBP2 in Figure 2.

Q4) There does not appear to be an inverse correlation in the expression of NRBP1 and NRBP2, according to Extended Data Figure 1a. Please use an inducible expression method or dose-dependency to prove each paralog's inhibitory role.

Q5) Have you ever treated other inhibitors for protein degradation rather than MG132? An example of control might be a lysosomal degradation inhibitor.

Q6) Please verify the alternative technique, such as ER extraction, in Figure 3a to validate the organelle localization of two paralogs.

Q7) Does the inverse correlation of NRBP1 and NRBP2 appear to be an expression pattern when using at least several cell lines?

Q8) Could you identify, in the complex purification data, the E3 ligases of NRBP1 via NRBP2? Have you ever used the poly-ubiquitination assay to verify the proteasomal degradation of NRBP1 by NRBP2?

Q9) HeLa cells were used by the authors to acquire RNA-seq data, but are these the most suitable cells to verify the inflammatory response?

Q10) If feasible, please confirm the L1 expression in Figure 6f using both healthy and RA patients.

Q11) What are the amino acid sequences for the NRBP invertebrates? Please conduct a direct comparison with human NRBP1 and NRBP2.

<Minor>

- The immunoblot in Extended Figure 1a does not match the text's description.
- Figure 3e (biological triplicate?) doesn't appear to be significant.

Version 1:

Reviewer comments:

Reviewer #1

(Remarks to the Author)

Yang et al. have gone to considerable efforts to satisfy my own concerns with their first submission, including multiple new experiments and new figure panels and text (and a new title and two new authors). I am satisfied that sufficient new work has been performed so the paper may proceed to publication.

This paper had four reviewers plus one co-reviewer. For that the authors have my sympathy.

Reviewer #2

(Remarks to the Author)

The authors have done a thorough job of addressing our concerns. We do request that the authors acknowledge explicitly the reason for including 7 biological replicates in Figure 1k, in the manuscript or in the Figure 1 legend.

Reviewer #3

(Remarks to the Author)

Reviewer #4

(Remarks to the Author)

I congratulate the authors for the additional experimental work added to the manuscript and the rebuttal letter for the comments of all the referees and I am very happy to consider the revised manuscript worth for publication.

Reviewer #5

(Remarks to the Author)

The authors have diligently addressed all points raised in the previous review round, providing substantial new data and revising the text and figures accordingly. I am very pleased with the revised version of the manuscript.

The manuscript is now significantly improved and presents a compelling story regarding the opposing roles of NRBP1 and NRBP2 in regulating L1 retrotransposition. I believe the work is now ready for publication, with only a few very minor points for the authors to consider for further clarity and precision in the final version.

My remaining suggestions are minor and pertain mostly to fine-tuning the language in the discussion to accurately reflect the scope of the findings:

Reciprocal Regulation and Inverse Correlation: The authors have thoroughly investigated the interplay between NRBP1 and NRBP2 expression and provided data that show some reciprocal effects, while also indicating variability or a lack of a strictly inverse correlation under all tested conditions (e.g., Extended Data Fig. 1a, 5e-g and related text). The discussion acknowledges this. I suggest ensuring the language throughout the manuscript precisely reflects the observed data, perhaps subtly emphasizing that the strong inhibitory effect of NRBP2 on NRBP1 protein levels is most consistently seen upon overexpression, or that the endogenous reciprocal regulation might be context-dependent, as indicated by their findings. This is a very minor point about wording precision.

Mechanism of NRBP1 Degradation: The authors have provided strong evidence ruling out classical proteasomal, lysosomal, and calpain-mediated degradation for NRBP1 induced by NRBP2 (Fig. 5 and related extended data). This is a valuable negative finding. While the precise alternative mechanism remains an exciting question for future research, the authors have clearly stated this in the discussion. No changes are needed here, other than perhaps ensuring the discussion clearly presents the identification of the specific degradation pathway as a significant future direction.

Link to Rheumatoid Arthritis Pathogenesis: Including the correlational data on NRBP2 mRNA levels in RA patients from a public database (Fig. 6f) is a useful addition that supports the potential clinical relevance. The authors have also appropriately explained why direct experimental validation in patient samples was not feasible currently. I suggest maintaining cautious language when discussing the link between NRBP2 and RA, framing it as a strong correlation and a compelling area for future investigation in clinically relevant samples, rather than a definitive causal link established in this study.

Inflammatory Response Model: The use of HeLa cells for inflammatory response analysis is understandable given the focus on L1 retrotransposition studies in this cell line. The authors have also acknowledged in the discussion the need for immune cells and patient samples for systematic investigation of NRBP2's role in inflammatory diseases. A brief reinforcing note in the results or discussion could simply reiterate that the inflammatory findings in HeLa cells suggest a potential link that warrants further validation in more complex biological systems.

The authors have successfully addressed the previous major concerns and the manuscript is now in excellent shape. The few points mentioned above are very minor suggestions for refinement. I recommend acceptance of the manuscript after these minimal revisions are considered.
